# Lifting Weak Supervision To Structured Prediction

**Harit Vishwakarma**
hvishwakarma@wisc.edu

**Frederic Sala**
fredsala@cs.wisc.edu

Department of Computer Sciences,
University of Wisconsin-Madison, WI, USA.

## Abstract

Weak supervision (WS) is a rich set of techniques that produce pseudolabels by aggregating easily obtained but potentially noisy label estimates from a variety of sources. WS is theoretically well understood for binary classification, where simple approaches enable consistent estimation of pseudolabel noise rates. Using this result, it has been shown that downstream models trained on the pseudolabels have generalization guarantees nearly identical to those trained on clean labels. While this is exciting, users often wish to use WS for *structured prediction*, where the output space consists of more than a binary or multi-class label set: e.g. rankings, graphs, manifolds, and more. Do the favorable theoretical properties of WS for binary classification lift to this setting? We answer this question in the affirmative for a wide range of scenarios. For labels taking values in a finite metric space, we introduce techniques new to weak supervision based on pseudo-Euclidean embeddings and tensor decompositions, providing a nearly-consistent noise rate estimator. For labels in constant-curvature Riemannian manifolds, we introduce new invariants that also yield consistent noise rate estimation. In both cases, when using the resulting pseudolabels in concert with a flexible downstream model, we obtain generalization guarantees nearly identical to those for models trained on clean data. Several of our results, which can be viewed as robustness guarantees in structured prediction with noisy labels, may be of independent interest. Empirical evaluation validates our claims and shows the merits of the proposed method[1].

## 1 Introduction

Weak supervision (WS) is an array of methods used to construct pseudolabels for training supervised models in label-constrained settings. The standard workflow [RSW+16, RBE+18, FCS+20] is to assemble a set of cheaply-acquired labeling functions—simple heuristics, small programs, pretrained models, knowledge base lookups—that produce multiple noisy estimates of what the true label is for each unlabeled point in a training set. These noisy outputs are modeled and aggregated into a single higher-quality pseudolabel. Any conventional supervised end model can be trained on these pseudolabels. This pattern has been used to deliver excellent performance in a range of domains in both research and industry settings [DRS+20, RNGS20, SLB20], bypassing the need to invest in large-scale manual labeling. Importantly, these successes are usually found in binary or small-cardinality classification settings.

While exciting, users often wish to use weak supervision in *structured prediction* (SP) settings, where the output space consists of more than a binary or multiclass label set [BHS+07, KL15]. In such cases, there exists meaningful algebraic or geometric structure to exploit. Structured prediction includes, for example, learning rankings used for recommendation systems [KAG18], regression in metric spaces [PM19], learning on manifolds [RCMR18], graph-based learning [GS19], and more.

---

[1] https://github.com/SprocketLab/WS-Struct-Pred

An important advantage of WS in the standard setting of binary classification is that it sometimes yields models with nearly the same generalization guarantees as their fully-supervised counterparts. Indeed, the penalty for using pseudolabels instead of clean labels is only a multiplicative constant. This is a highly favorable tradeoff since acquiring more unlabeled data is easy. This property leads us to ask the key question for this work: **does weak supervision for structured prediction preserve generalization guarantees?** We answer this question in the affirmative, justifying the application of WS to settings far from its current use.

Generalization results in WS rely on two steps [RHD+19, FCS+20]: (i) showing that the estimator used to learn the model of the labeling functions is consistent, thus recovering the noise rates for these noisy voters, and (ii) using a noise-aware loss to de-bias end-model training [NDRT13]. Lifting these two results to structured prediction is challenging. The only available weak supervision technique suitable for SP is that of [SLV+22]. It suffers from several limitations. First, it relies on the availability of isometric embeddings of metric spaces into $\mathbb{R}^d$—but does not explain how to find these. Second, it does not tackle downstream generalization at all. We resolve these two challenges.

We introduce results for a wide variety of structured prediction problems, requiring only that the labels live in some metric space. We consider both finite and continuous (manifold-valued) settings. For finite spaces, we apply two tools that are new to weak supervision. The approach we propose combines isometric *pseudo-Euclidean embeddings* with *tensor decompositions*—resulting in a nearly-consistent noise rate estimator. In the continuous case, we introduce a label model suitable for the so-called *model spaces*—Riemannian manifolds of constant curvature—along with extensions to even more general spaces. In both cases, we show generalization results when using the resulting pseudolabels in concert with a flexible end model from [CRR16, RCMR18].

**Contributions:**

- New techniques for performing weak supervision in finite metric spaces based on isometric pseudo-Euclidean embeddings and tensor decomposition algorithms,
- Generalizations to manifold-valued regression in constant-curvature manifolds,
- Finite-sample error bounds for noise rate estimation in each scenario,
- Generalization error guarantees for training downstream models on pseudolabels,
- Experiments confirming the theoretical results and showing improvements over [SLV+22].

## 2 Background and Problem Setup

Our goal is to theoretically characterize how well learning with pseudolabels (built with weak supervision techniques) performs in structured prediction. We seek to understand the interplay between the noise in WS sources and the generalization performance of the downstream structured prediction model. We provide brief background and introduce our problem and some useful notation.

### 2.1 Structured Prediction

Structured prediction (SP) involves predicting labels in spaces with rich structure. Denote the label space by $\mathcal{Y}$. Conventionally $\mathcal{Y}$ is a set, e.g., $\mathcal{Y} = \{-1, +1\}$ for binary classification. In the SP setting, $\mathcal{Y}$ has some additional algebraic or geometric structure. In this work we assume that $\mathcal{Y}$ is a metric space with metric (distance) $d_{\mathcal{Y}}$. This covers many types of problems, including

- Rankings, where $\mathcal{Y} = S_\rho$, the symmetric group on $\{1, \ldots, \rho\}$, i.e., labels are permutations,
- Graphs, where $\mathcal{Y} = \mathcal{G}_\rho$, the space of graphs with vertex set $V = \{1, \ldots, \rho\}$,
- Riemannian manifolds, including $\mathcal{Y} = \mathbb{S}_d$, the sphere, or $\mathbb{H}_d$, the hyperboloid.

**Learning and Generalization in Structured Prediction**  In conventional supervised learning we have a dataset $\{(x_1, y_1), \ldots, (x_n, y_n)\}$ of i.i.d. samples drawn from distribution $\rho$ over $\mathcal{X} \times \mathcal{Y}$. As usual, we seek to learn a model that generalizes well to points not seen during training. Let $\mathcal{F} = \{f : \mathcal{X} \mapsto \mathcal{Y}\}$ be a family of functions from $\mathcal{X}$ to $\mathcal{Y}$. Define the risk $R(f)$ for $f \in \mathcal{F}$ and $f^*$ as

$$R(f) = \int_{\mathcal{X} \times \mathcal{Y}} d_{\mathcal{Y}}^2(f(x), y) d\rho(x, y) \qquad f^* \in \arg\min_{f \in \mathcal{F}} R(f). \qquad (1)$$

For a large class of settings (including all of those we consider in this paper), [CRR16, RCMR18] have shown that the estimator $\hat{f}$ is consistent:

$$\hat{f}(x) = \arg\min_{y \in \mathcal{Y}} F(x, y) \qquad F(x, y) := \frac{1}{n} \sum_{i=1}^{n} \alpha_i(x) d_{\mathcal{Y}}^2(y, y_i), \qquad (2)$$

where $\alpha(x) = (\mathbf{K} + \nu \mathbf{I})^{-1} \mathbf{K}_x$. Here, $\mathbf{K}$ is the kernel matrix for a p.d. kernel $k : \mathcal{X} \times \mathcal{X} \to \mathbb{R}$, so that $\mathbf{K}_{i,j} = k(x_i, x_j)$, $(\mathbf{K}_x)_i = k(x, x_i)$, and $\nu$ is a regularization parameter. The procedure here is to first compute the weights $\alpha$ and then to perform the optimization in (2) to make a prediction.

An exciting contribution of [CRR16, RCMR18] is the generalization bound

$$R(\hat{f}) \leq R(f^*) + \mathcal{O}(n^{-\frac{1}{4}}),$$

that holds with high probability, as long as there is no label noise. The key question we tackle is *does the use of pseudolabels instead of true labels $y_i$ affect the generalization rate?*

## 2.2 Weak Supervision

In WS, we cannot access *any* of the ground-truth labels $y_i$. Instead we observe for each $x_i$ the noisy votes $\lambda_{1,i}, \dots, \lambda_{m,i}$. These are $m$ weak supervision outputs provided by *labeling functions* (LFs) $s_a$, where $s_a : \mathcal{X} \to \mathcal{Y}$ and $\lambda_{a,i} = s_a(x_i)$. A two step process is used to construct pseudolabels. First, we learn a *noise model* (also called a label model) that determines how reliable each source $s_a$ is. That is, we must learn $\boldsymbol{\theta}$ for $P_{\boldsymbol{\theta}}(\lambda_1, \lambda_2, \dots, \lambda_m | y)$—without having access to any samples of $y$. Second, the noise model is used to infer a distribution (or its mode) for each point: $P_{\boldsymbol{\theta}}(y_i | \lambda_{1,i}, \dots, \lambda_{m,i})$.

We adopt the noise model from [SLV+22], which is suitable for our SP setting:

$$P_{\boldsymbol{\theta}}(\lambda_1, \dots, \lambda_m | Y = y) = \frac{1}{Z} \exp\left( -\sum_{a=1}^{m} \theta_a d_{\mathcal{Y}}^2(\lambda_a, y) - \sum_{(a,b) \in E} \theta_{a,b} d_{\mathcal{Y}}^2(\lambda_a, \lambda_b) \right). \qquad (3)$$

$Z$ is the normalizing partition function, $\boldsymbol{\theta} = [\theta_1, \dots, \theta_m]^T > 0$ are *canonical* parameters, and $E$ is a set of correlations. The model can be described in terms of the *mean* parameters $\mathbb{E}[d_{\mathcal{Y}}^2(\lambda_a, y)]$. Intuitively, if $\theta_a$ is large, the typical distance from $\lambda_a$ to $y$ is small and the LF is reliable; if $\theta_a$ is small, the LF is unreliable. This model is appropriate for several reasons. It is an exponential family model with useful theoretical properties. It subsumes popular special cases of noise, including, for regression, zero-mean multivariate Gaussian noise; for permutations, a generalization of the popular Mallows model; for the binary case, it produces a close relative of the Ising model.

Our goal is to form estimates $\hat{\boldsymbol{\theta}}$ in order to construct pseudolabels. One way to build such pseudolabels is to compute $\tilde{y} = \arg\min_{z \in \mathcal{Y}} 1/m \sum_{a=1}^{m} \hat{\theta}_a d_{\mathcal{Y}}^2(z, \lambda_a)$. Observe how the estimated parameters $\hat{\theta}_a$ are used to weight the labeling functions, ensuring that more reliable votes receive a larger weight.

We are now in a position to state the main research question for this work:

**Do there exist estimation approaches yielding $\hat{\boldsymbol{\theta}}$ that produce pseudolabels $\tilde{y}$ that maintain the same generalization error rate $\mathcal{O}(n^{-1/4})$ when used in (2), or a modified version of (2)?**

## 3 Noise Rate Recovery in Finite Metric Spaces

In the next two sections we handle finite metric spaces. Afterwards we tackle continuous (manifold-valued) spaces. We first discuss learning the noise parameters $\boldsymbol{\theta}$, then the use of pseudolabels.

**Roadmap** For finite metric spaces with $|\mathcal{Y}| = r$, we apply two tools new to weak supervision. First, we embed $\mathcal{Y}$ into a *pseudo-Euclidean* space [Gol85]. These spaces generalize Euclidean space, enabling isometric (distance-preserving) embeddings for any metric. Using pseudo-Euclidean spaces make our analysis slightly more complex, but we gain the isometry property, which is critical.

Second, we form three-way tensors from embeddings of observed labeling functions. Applying tensor product decomposition algorithms [AGH+14], we can recover estimates of the mean parameters $\hat{\mathbb{E}}[d_{\mathcal{Y}}^2(\lambda_a, y)]$ and ultimately $\hat{\theta}_a$. Finally, we reweight the model (2) to preserve generalization.

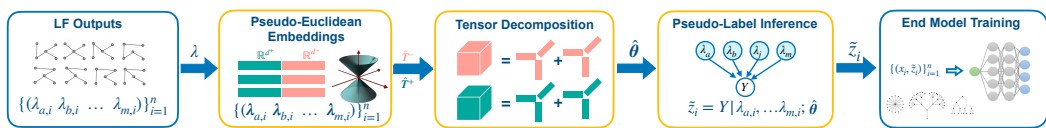

Figure 1: Illustration of our weak supervision pipeline for the finite label space setting.

The intuition behind this approach is the following. First, we need a technique that can provide consistent or nearly-consistent estimates of the parameters in the noise model. Second, we need to handle any finite metric space. Techniques like the one introduced in [FCS$^+$20] handle the first—but do not work for generic finite metric spaces, only binary labels and certain sequences. Techniques like the one in [SLV$^+$22] handle any metric space—but only have consistency guarantees in highly restrictive settings (e.g., it requires an isometric embedding, that the distribution over the resulting embeddings is isomorphic to certain distributions, the true label only takes on two values). Pseudo-Euclidean embeddings used with tensor decomposition algorithms meet both requirements

### 3.1 Pseudo-Euclidean Embeddings

Our first task is to embed the metric space into a continuous space—enabling easier computation and potential dimensionality reduction. A standard approach is multi-dimensional scaling (MDS) [KW78], which embeds $\mathcal{Y}$ into $\mathbb{R}^d$. A downside of MDS is that not all metric spaces embed (isometrically) into Euclidean space, as the square distance matrix $\mathbf{D}$ must be positive semi-definite.

A simple and elegant way to overcome this difficulty is to instead use *pseudo-Euclidean* spaces for embeddings. These pseudo-spaces do not require a p.s.d. inner product. As an outcome, any finite metric space can be embedded into a pseudo-Euclidean space with *no distortion* [Gol85]—so that distances are exactly preserved. Such spaces have been applied to similarity-based learning methods [PPD01, LRBM06, PHD$^+$06]. A vector $\mathbf{u}$ in a pseudo-Euclidean space $\mathbb{R}^{d^+,d^-}$ has two parts: $\mathbf{u}^+ \in \mathbb{R}^{d^+}$ and $\mathbf{u}^- \in \mathbb{R}^{d^-}$. The dot product and the squared distance between any two vectors $\mathbf{u}, \mathbf{v}$ are $\langle \mathbf{u}, \mathbf{v} \rangle_\phi = \langle \mathbf{u}^+, \mathbf{v}^+ \rangle - \langle \mathbf{u}^-, \mathbf{v}^- \rangle$ and $d_\phi^2(\mathbf{u}, \mathbf{v}) = ||\mathbf{u}^+ - \mathbf{v}^+||_2^2 - ||\mathbf{u}^- - \mathbf{v}^-||_2^2$. These properties enable isometric embeddings: the distance can be decomposed into two components that are individually induced from p.s.d. inner products—and can thus be embedded via MDS. Indeed, pseudo-Euclidean embeddings effectively run MDS for each component (see Algorithm 1 steps 4-9). To recover the original distance, we obtain $||\mathbf{u}^+ - \mathbf{v}^+||_2^2$ and $||\mathbf{u}^- - \mathbf{v}^-||_2^2$ and subtract.

*Example:* To see why such embeddings are advantageous, we compare with a one-hot vector representation (whose dimension is $|\mathcal{Y}|$). Consider a tree with a root node and three branches, each of which is a path with $t$ nodes. Let $\mathcal{Y}$ be the nodes in the tree with the shortest-hops distance as the metric. The pseudo-Euclidean embedding dimension is just $d = 3$; see Appendix for more details. The one-hot embedding dimension is $d = |\mathcal{Y}| = 3t + 1$—arbitrarily larger!

Now we are ready to apply these embeddings to our problem. Abusing notation, we write $\boldsymbol{\lambda}_a$ and $\mathbf{y}$ for the pseudo-Euclidean embeddings of $\lambda_a, y$, respectively. We have that $d_\mathcal{Y}^2(\lambda_a, y) = d_\phi^2(\boldsymbol{\lambda}_a, \mathbf{y})$, so that there is no loss of information from working with these spaces. In addition, we write the mean as $\boldsymbol{\mu}_{a,y} = \mathbb{E}[\boldsymbol{\lambda}_a | \mathbf{y}]$ and the covariance as $\boldsymbol{\Sigma}_{a,y}$. Our goal is to obtain an accurate estimate $\hat{\boldsymbol{\mu}}_{a,y} = \hat{\mathbb{E}}[\boldsymbol{\lambda}_a | \mathbf{y}]$, which we will use to estimate the mean parameters $\mathbb{E}[d_\mathcal{Y}^2(\lambda_a, y)]$. If we could observe $y$, it would be easy to empirically estimate $\boldsymbol{\mu}_{a,y}$—but we do not have access to it. Our approach will be to apply tensor decomposition for multi-view mixtures [AGJ14].

### 3.2 Multi-View Mixtures and Tensor Decompositions

In a multi-view mixture model, multiple views $\{\lambda_a\}_{a=1}^m$ of a latent variable $Y$ are observed. These views are independent when conditioned on $Y$. We treat the positive and negative components $\boldsymbol{\lambda}_a^+ \in \mathbb{R}^{d^+}$ and $\boldsymbol{\lambda}_a^- \in \mathbb{R}^{d^-}$ of our pseudo-Euclidean embedding as separate multi-view mixtures:

$$\boldsymbol{\lambda}_a^+ | \mathbf{y} \sim \boldsymbol{\mu}_{a,y}^+ + \sigma\sqrt{d^+} \cdot \boldsymbol{\epsilon}_a^+ \quad \text{and} \quad \boldsymbol{\lambda}_a^- | \mathbf{y} \sim \boldsymbol{\mu}_{a,y}^- + \sigma\sqrt{d^-} \cdot \boldsymbol{\epsilon}_a^- \qquad \forall a \in [m], \qquad (4)$$

where $\boldsymbol{\mu}_{a,y}^+ = \mathbb{E}[\boldsymbol{\lambda}_a^+ | \mathbf{y}]$, $\boldsymbol{\mu}_{a,y}^- = \mathbb{E}[\boldsymbol{\lambda}_a^- | \mathbf{y}]$ and $\boldsymbol{\epsilon}_a^+, \boldsymbol{\epsilon}_a^-$ are mean zero random vectors with covariances $\frac{1}{d^+}\mathbf{I}_{d^+}, \frac{1}{d^-}\mathbf{I}_{d^-}$ respectively. Here $\sigma^2$ is a proxy variance whose use is described in Assumption 3.

---
**Algorithm 1** Algorithm for Pseudolabel Construction
---
**Input:** Labeling function outputs $\mathbf{L} = \{(\lambda_{1,i}, \ldots, \lambda_{m,i})\}_{i=1}^n$, Label Space $\mathcal{Y} = \{y_0, \ldots, y_{r-1}\}$
**Output:** Pseudolabels for each data point $\mathbf{Z} = \{\tilde{z}_i\}_{i=1}^n$

    ▷ Step 1: Compute pseudo-Euclidean Embeddings
1: Construct matrices $\mathbf{D} \in \mathbb{R}^{r \times r}$, $\mathbf{D}_{ij} = d_{\mathcal{Y}}^2(y_i, y_j)$ and $\mathbf{M} \in \mathbb{R}^{r \times r}$, $\mathbf{M}_{ij} = \frac{1}{2}(\mathbf{D}_{0i}^2 + \mathbf{D}_{0j}^2 - \mathbf{D}_{ij}^2)$
2: Compute eigendecomposition of $\mathbf{M}$ and let $\mathbf{M} = \mathbf{U}\mathbf{C}\mathbf{U}^T$
3: Set $l^+, l^-$ be indices of positive and negative eigenvalues sorted by their magnitude
4: Let $d^+ = |l^+|, \quad d^- = |l^-|$ i.e. the sizes of lists $l^+$ and $l^-$ respectively.
5: Construct permutation matrix $\mathbf{I}_{perm} \in \mathbb{R}^{r \times (d^+ + d^-)}$ by concatenating $l^+, l^-$ in order
6: $\bar{\mathbf{C}} = \mathbf{C}\mathbf{I}_{perm}, \bar{\mathbf{U}} = \mathbf{U}\mathbf{I}_{perm}$
7: $\mathbb{Y} = \bar{\mathbf{U}}^T\bar{\mathbf{C}}^{\frac{1}{2}} \in \mathbb{R}^{r \times (d^+ + d^-)}$ and let this define the mapping $g : \mathcal{Y} \mapsto \mathbb{Y}$

    ▷ Step 2: Parameter Estimation Using Tensor Decomposition
8: **for** $a \leftarrow 1$ to $m - 3$ **do**
9:     Obtain embeddings $\boldsymbol{\lambda}_{a,i} = g(\lambda_{a,i}), \boldsymbol{\lambda}_{b,i} = g(\lambda_{b,i}), \boldsymbol{\lambda}_{c,i} = g(\lambda_{c,i}) \quad \forall i \in [n]$ where $a, b, c$ are uncorrelated
10:     Construct tensors $\hat{\mathbf{T}}^+$ and $\hat{\mathbf{T}}^-$ as defined in (5) for triplet $(a, b, c)$
11:     $\hat{\boldsymbol{\mu}}_{a,y}^+, \hat{\boldsymbol{\mu}}_{b,y}^+, \hat{\boldsymbol{\mu}}_{c,y}^+ = \texttt{TensorDecomposition}(\hat{\mathbf{T}}^+)$
12:     $\hat{\boldsymbol{\mu}}_{a,y}^-, \hat{\boldsymbol{\mu}}_{b,y}^-, \hat{\boldsymbol{\mu}}_{c,y}^- = \texttt{TensorDecomposition}(\hat{\mathbf{T}}^-)$
13:     $s_{a,y}^+ = \min_{z \in \{-1,+1\}} \phi(z \cdot \hat{\boldsymbol{\mu}}_{a,y}^+, \mathbf{y}^+)$ and similarly $s_{b,y}^+, s_{c,y}^+, s_{a,y}^-, s_{b,y}^-, s_{c,y}^-$
14:     $\hat{\boldsymbol{\mu}}_{a,y}^+ = s_{a,y}^+ \cdot \hat{\boldsymbol{\mu}}_{a,y}^+$ and similarly correct signs of $\hat{\boldsymbol{\mu}}_{b,y}^+, \hat{\boldsymbol{\mu}}_{c,y}^+, \hat{\boldsymbol{\mu}}_{a,y}^-, \hat{\boldsymbol{\mu}}_{b,y}^-, \hat{\boldsymbol{\mu}}_{c,y}^-$
15: **end for**

    ▷ Step 3: Infer Pseudo-Labels
16: $\tilde{Z}^{(i)} = \tilde{z}_i \sim Y | \lambda_a = \lambda_a^{(i)}, \ldots \lambda_m = \lambda_m^{(i)}; \hat{\boldsymbol{\theta}}$

17: **return** $\{\tilde{z}_i\}_{i=1}^n$

---

We cannot directly estimate these parameters from observations of $\boldsymbol{\lambda}_a$, due to the fact that $\mathbf{y}$ is not observed. However, we can observe various moments of the outputs of the LFs such as tensors of outer products of LF triplets. We require that for each $a$ such a triplet exists. Then,

$$\mathbf{T}^+ := \mathbb{E}[\boldsymbol{\lambda}_a^+ \otimes \boldsymbol{\lambda}_b^+ \otimes \boldsymbol{\lambda}_c^+] = \sum_{y \in \mathcal{Y}_s} w_y \boldsymbol{\mu}_{a,y}^+ \otimes \boldsymbol{\mu}_{b,y}^+ \otimes \boldsymbol{\mu}_{c,y}^+ \text{ and } \hat{\mathbf{T}}^+ := \frac{1}{n} \sum_{i=1}^n \boldsymbol{\lambda}_{a,i}^+ \otimes \boldsymbol{\lambda}_{b,i}^+ \otimes \boldsymbol{\lambda}_{c,i}^+. \quad (5)$$

Here $w_y$ are the mixture probabilities (prior probabilities of $Y$) and $\mathcal{Y}_s = \{y : w_y > 0\}$. We similarly define $\mathbf{T}^-$ and $\hat{\mathbf{T}}^-$. We then obtain estimates $\hat{\boldsymbol{\mu}}_{a,y}^+, \hat{\boldsymbol{\mu}}_{a,y}^-$ using an algorithm from [AGH+14] with minor modifications to handle pseudo-Euclidean rather than Euclidean space. The overall approach is shown in Algorithm 1. We have three key assumptions for our analysis,

**Assumption 1.** *The support of $P_Y$, i.e., $k = |\{y : w_y > 0\}|$ and the label space $\mathcal{Y}$ is such that $\min(d^+, d^-) \geq k$, $\|\boldsymbol{\mu}_{a,y}^+\|_2 = 1, \|\boldsymbol{\mu}_{a,y}^-\|_2 = 1$ for $a \in [m], y \in \mathcal{Y}$.*

**Assumption 2.** *(Bounded angle between $\boldsymbol{\mu}$ and $\mathbf{y}$) Let $\phi(\mathbf{u}, \mathbf{v})$ denote the angle between any two vectors $\mathbf{u}, \mathbf{v}$ in a Euclidean space. We assume that $\phi(\boldsymbol{\mu}_{a,y}^+, \mathbf{y}^+) \in [0, \pi/2 - c)$, $\phi(\boldsymbol{\mu}_{a,y}^-, \mathbf{y}^-) \in [0, \pi/2 - c) \forall a \in [m]$, and $y \in \mathcal{Y}_s$, for some sufficiently small $c \in (0, \pi/4)$ such that $\sin(c) \geq \max(\epsilon_0(d^+), \epsilon_0(d^-))$, where $\epsilon_0(d)$ is defined for some $n > n_0$ samples in (6).*

**Assumption 3.** *$\sigma$ is such that the recovery error with model (4) is at least as large as with (3).*

These enable providing guarantees on recovering the mean vector magnitudes (1) and signs (2) and simplify the analysis (1), (3); all three can be relaxed at the expense of a more complex analysis.

Our first theoretical result shows that we have near-consistency in estimating the mean parameters in (3). We use standard notation $\tilde{\mathcal{O}}$ ignoring logarithmic factors.

**Theorem 1.** *Let $\hat{\boldsymbol{\mu}}_{a,y}^+, \hat{\boldsymbol{\mu}}_{a,y}^-$ be the estimates of $\boldsymbol{\mu}_{a,y}^+, \boldsymbol{\mu}_{a,y}^-$ returned by Algorithm 1 with input $\hat{\mathbf{T}}^+, \hat{\mathbf{T}}^-$ constructed using isometric pseudo-Euclidean embeddings (in $\mathbb{R}^{d^+, d^-}$). Suppose Assumptions 1 and 2 are met, a sufficiently large number of samples $n$ are drawn from the model in (3), and $k = |\mathcal{Y}_s|$. Then there exists a constant $C_0 > 0$ such that with high probability $\forall a \in [m]$ and $y \in \mathcal{Y}_s$,*

$$|\theta_a - \hat{\theta}_a| \leq C_0 \left| \mathbb{E}[d_{\mathcal{Y}}^2(\lambda_a, y)] - \hat{\mathbb{E}}[d_{\mathcal{Y}}^2(\lambda_a, y)] \right| \leq \epsilon(d^+) + \epsilon(d^-),$$

*where*

$$\epsilon(d) := \begin{cases} \tilde{\mathcal{O}}\left(k\sqrt{\frac{d}{n}}\right) + \tilde{\mathcal{O}}\left(\frac{\sqrt{k}}{d}\right) & \text{if } \sigma^2 = \Theta(1), \\ \tilde{\mathcal{O}}\left(\sqrt{\frac{k}{n}}\right) + \tilde{\mathcal{O}}\left(\frac{\sqrt{k}}{d}\right) & \text{if } \sigma^2 = \Theta(\frac{1}{d}). \end{cases} \tag{6}$$

We interpret Theorem 1. It is a nearly direct application of [AGJ14]. There are two noise cases for $\sigma$. In the high-noise case, $\sigma$ is independent of dimension $d$ (and thus $|\mathcal{Y}|$). Intuitively, this means the average distance balls around each LF begin to overlap as the number of points grows—explaining the multiplicative $k$ term. If the noise scales down as we add more embedded points, this problem is removed, as in the low-noise case. In both cases, the second error term comes from using the algorithm of [AGH+14] and is independent of the sampling error. Since $k = \Theta(d)$, this term goes down with $d$. The first error term is due to sampling noise and goes to zero in the number of samples $n$. Note the tradeoffs of using the embeddings. If we used one-hot encoding, $d = |\mathcal{Y}|$, and in the high-noise case, we would pay a very heavy cost for $\sqrt{d/n}$. However, while sampling error is minimized when using a very small $d$, we pay a cost in the second error term. This leads to a tradeoff in selecting the appropriate embedding dimension.

## 4 Generalization Error for Structured Prediction in Finite Metric Spaces

We have access to labeling function outputs $\lambda_{a,i}, \ldots, \lambda_{m,i}$ for points $x_i$ and noise rate estimates $\hat{\theta}_a, \ldots, \hat{\theta}_m$. How can we use these to infer unobserved labels $y$ in (2)? Our approach is based on [NDRT13, vRW18], where the underlying loss function is modified to deal with noise. Analogously, we modify (2) in such a way that the generalization guarantee is nearly preserved.

### 4.1 Prediction with Pseudolabels

First, we construct the posterior distribution $P_{\hat{\boldsymbol{\theta}}}(Y = y|\lambda)$. We use our estimated noise model $P_{\hat{\boldsymbol{\theta}}}(\lambda|Y)$ and the prior $P(Y = y)$. We create pseudo-labels for each data point by drawing a random sample from the posterior distribution conditioned on the output of labeling functions: $\tilde{Z}^{(i)} = \tilde{z}_i \sim Y|\lambda_a = \lambda_a^{(i)}, \ldots, \lambda_m = \lambda_m^{(i)}; \hat{\boldsymbol{\theta}}$. We thus observe $(x_1, \tilde{z}_1), \ldots, (x_n, \tilde{z}_n)$ where $\tilde{z}_i$ is sampled as above. To overcome the effect of noise we create a perturbed version of the distance function using the noise rates, generalizing [NDRT13]. This requires us to characterize the noise distribution induced by our inference procedure. In particular we seek the probability that $\tilde{Z} = y_j$ when the true label is $y_j$. This can be expressed as follows. Let $\mathcal{Y}^m$ denote the $m$-fold Cartesian product of $\mathcal{Y}$ and let $\Lambda_u = (\lambda_1^{(u)}, \ldots, \lambda_m^{(u)})$ denote its $u^{th}$ entry. We write

$$\mathbf{P}_{ij} = P_{\boldsymbol{\theta}}(\tilde{Z} = y_j|Y = y_i) = \sum_{u=1}^{|\mathcal{Y}^m|} P_{\boldsymbol{\theta}}(Y = y_j|\Lambda = \Lambda^{(u)}) \cdot P_{\boldsymbol{\theta}}(\Lambda = \Lambda^{(u)}|Y = y_i). \tag{7}$$

We define $\mathbf{Q}_{ij} = P_{\hat{\boldsymbol{\theta}}}(\tilde{Z} = y_j|Y = y_i)$ using $\hat{\boldsymbol{\theta}}$. $\mathbf{P}$ is the noise distribution induced by the true parameters $\boldsymbol{\theta}$ and $\mathbf{Q}$ is an approximation obtained from inference with the *estimated* parameters $\hat{\boldsymbol{\theta}}$. With this terminology, we can define the perturbed version of the distance function and a corresponding replacement of (2):

$$\tilde{d}_q(T, \tilde{Y} = y_j) := \sum_{i=1}^k (\mathbf{Q}^{-1})_{ji} d_{\mathcal{Y}}^2(T, Y = y_i) \quad \forall y_j \in \mathcal{Y}, \tag{8}$$

$$\tilde{F}_q(x, y) := \frac{1}{n} \sum_{i=1}^n \alpha_i(x) \tilde{d}_q(y, \tilde{z}_i) \qquad \hat{f}_q(x) = \arg\min_{y \in \mathcal{Y}} \tilde{F}_q(x, y). \tag{9}$$

We similarly define $\tilde{d}_p, \tilde{F}_p, \hat{f}_p$ using the true noise distribution $\mathbf{P}$. The perturbed distance $\tilde{d}_p$ is an unbiased estimator of the true distance. However we do not know the true noise distribution $\mathbf{P}$ hence we cannot use it for prediction. Instead we use $\hat{d}_q$. Note that $\hat{d}_q$ is no longer an unbiased estimator—its bias can be expressed as function of the parameter recovery error bound in Theorem 1.

## 4.2 Bounding the Generalization Error

What can we say about the excess risk $R(\hat{f}_q) - R(f^*)$? Note that compared to the prediction based on clean labels, there are two additional sources of error. One is the noise in the labels (i.e., even if we know the true $\mathbf{P}$, the quality of the pseudolabels is imperfect). The other is our estimation procedure for the noise distribution. We must address both sources of error.

Our analysis uses the following assumptions on the minimum and maximum singular values $\sigma_{\min}(\mathbf{P})$ , $\sigma_{\max}(\mathbf{P})$ and the condition number $\kappa(\mathbf{P})$ of true noise matrix $\mathbf{P}$ and the function $F$. Additional detail is provided in the Appendix.

**Assumption 4.** *(Noise model is not arbitrary) The true parameters $\boldsymbol{\theta}$ are such that $\sigma_{\min}(\mathbf{P}) > 0$, and the condition number $\kappa(\mathbf{P})$ is sufficiently small.*

**Assumption 5.** *(Normalized features) $|\alpha(x)| \leq 1$, for all $x \in \mathcal{X}$.*

**Assumption 6.** *(Proxy strong convexity) The function $F$ in (2) satisfies the following property with some $\beta > 0$. As we move away from the minimizer of $F$, the function increases and the rate of increase is proportional to the distance between the points:*

$$F\big(x, f(x)\big) \geq F\big(x, \hat{f}(x)\big) + \beta \cdot d_{\mathcal{Y}}^2\big(f(x), \hat{f}(x)\big) \qquad \forall x \in \mathcal{X}, \forall f \in \mathcal{F}. \tag{10}$$

With these assumptions, we provide a generalization result for prediction with pseudolabels,

**Theorem 2.** *(Generalization Error ) Let $\hat{f}$ be the minimizer as defined in (2) over the clean labels and let $\hat{f}_q$ (defined in (9)) be the minimizer over the noisy labels obtained from inference in Algorithm 1. Suppose Assumptions 4,5,6 hold. Then for $\epsilon_2 = k^{5/2} \cdot \tilde{\mathcal{O}}(\epsilon(d^+) + \epsilon(d^-)) \cdot \left(1 + \frac{\kappa(\mathbf{P})}{\sigma_{\min}(\mathbf{P})}\right)$ and $c_1 = 1 + \frac{\sqrt{k}}{\sigma_{\min}(\mathbf{P})}$, with high probability,*

$$R(\hat{f}_q) \leq R(f^*) + \mathcal{O}(n^{-\frac{1}{4}}) + \tilde{\mathcal{O}}\Big(\frac{c_1}{\beta} n^{-\frac{1}{2}}\Big) + \tilde{\mathcal{O}}\Big(\frac{3\epsilon_2}{\beta} n^{-\frac{1}{2}}\Big). \tag{11}$$

**Implications and Tradeoffs:** We interpret each term in the bound. The first term is present even with access to the clean labels and hence unavoidable. The second term is the additional error we incur if we learn with the knowledge of the true noise distribution. The third term is due to the use of the estimated noise model. It is dominated by the noise rate recovery result in Theorem 1. If the third term goes to 0 (perfect recovery) then we obtain the rate $\mathcal{O}(n^{-1/4})$, the same as in the case of access to clean labels. The third term is introduced by our noise rate recovery algorithm and has two terms: one dominated by $\tilde{\mathcal{O}}(n^{-1/2})$ and the other on $\tilde{\mathcal{O}}(\sqrt{k}/d)$ (see discussion of Theorem 1). Thus we only pay an extra additive factor $\mathcal{O}(\sqrt{k}/d)$ in the excess risk when using pseudolabels.

## 5 Manifold-Valued Label Spaces: Noise Recovery and Generalization

We introduce a simple recovery method for weak supervision in constant-curvature Riemannian manifolds. First we briefly introduce some background notation on these spaces, then provide our estimator and consistency result, then the downstream generalization result. Finally, we discuss extensions to symmetric Riemannian manifolds, an even more general class of spaces.

**Background on Riemannian manifolds** The following is necessarily a very abridged background; more detail can be found in [Lee00, Tu11]. A smooth manifold $M$ is a space where each point is located in a neighborhood diffeomorphic to $\mathbb{R}^d$. Attached to each point $p \in \mathcal{M}$ is a *tangent space* $T_p M$; each such tangent space is a $d$-dimensional vector space enabling the use of calculus.

A Riemannian manifold equips a smooth manifold with a Riemannian metric: a smoothly-varying inner product $\langle \cdot, \cdot \rangle_p$ at each point $p$. This tool allows us to compute angles, lengths, and ultimately, distances $d_{\mathcal{M}}(p, q)$ between points on the manifold as shortest-path distances. These shortest paths are called geodesics and can be parametrized as curves $\gamma(t)$, where $\gamma(0) = p$, or by tangent vectors $v \in T_p M$. The exponential map operation $\exp : T_p \mathcal{M} \mapsto \mathcal{M}$ takes tangent vectors to manifold points. It enables switching between these tangent vectors: $\exp_p(v) = q$ implies that $d_{\mathcal{M}}(p, q) = \|v\|$. The logarithmic map operation $\log : \mathcal{M} \mapsto T_p \mathcal{M}$ takes manifold points to tangent vectors. Further, $\exp_p(v) = q$ is equivalent to $\log_p(q) = v$.

**Invariant**   Our first contribution is a simple invariant that enables us to recover the error parameters. Note that we cannot rely on the finite metric-space technique, since the manifolds we consider have an infinite number of points. Nor do we need an embedding—we have a continuous representation as-is. Instead, we propose a simple idea based on the law of cosines. Essentially, on average, the geodesic triangle formed by the latent variable $y \in \mathcal{M}$ and two observed LFs $\lambda_a, \lambda_b$, is a right triangle. This means it can be characterized by the (Riemannian) version of the Pythagorean theorem:

**Lemma 1.** *For $\mathcal{Y} = \mathcal{M}$, a hyperbolic manifold, $y \sim P$ for some distribution $P$ on $\mathcal{M}$ and labeling functions $\lambda_a, \lambda_b$ drawn from (3), $\mathbb{E} \cosh d_{\mathcal{Y}}(\lambda_a, \lambda_b) = \mathbb{E} \cosh d_{\mathcal{Y}}(\lambda_b, y) \mathbb{E} \cosh d_{\mathcal{Y}}(\lambda_b, y)$, while for $\mathcal{Y} = \mathcal{M}$ a spherical manifold, $\mathbb{E} \cos d_{\mathcal{Y}}(\lambda_a, \lambda_b) = \mathbb{E} \cos d_{\mathcal{Y}}(\lambda_b, y) \mathbb{E} \cos d_{\mathcal{Y}}(\lambda_b, y)$.*

These invariants enable us to easily learn by forming a triplet system. Suppose we construct the equation in Lemma 1 for three pairs of labeling functions. The resulting system can be solved to express $\mathbb{E}[\cosh(d_{\mathcal{Y}}(\lambda_a, y))]$ in terms of $\mathbb{E} \cosh(d_{\mathcal{Y}}(\lambda_a, \lambda_b)), \mathbb{E} \cosh(d_{\mathcal{Y}}(\lambda_a, \lambda_c)), \mathbb{E} \cosh(d_{\mathcal{Y}}(\lambda_b, \lambda_c))$. Specifically,

$$\mathbb{E} \cosh(d_{\mathcal{Y}}(\lambda_a, y)) = \sqrt{\frac{\mathbb{E} \cosh d_{\mathcal{Y}}(\lambda_a, \lambda_b) \mathbb{E} \cosh d_{\mathcal{Y}}(\lambda_a, \lambda_c)}{(\mathbb{E} \cosh(d_{\mathcal{Y}}(\lambda_b, \lambda_c)))^2}}.$$

Note that we can estimate $\hat{\mathbb{E}}$ via the empirical versions of terms on the right , as these are based on observable quantities. This is a generalization of the binary case in [FCS$^+$20] and the Gaussian (Euclidean) case in [SLV$^+$22] to hyperbolic manifolds. A similar estimator can be obtained for spherical manifolds by replacing $\cosh$ with $\cos$.

Using this tool, we can obtain a consistent estimator for $\theta_a$ for each of $a = 1, \ldots, m$. Let $C_0$ satisfy $\mathbb{E}|\hat{\mathbb{E}} \cosh(d_{\mathcal{Y}}(\lambda_a, \lambda_b)) - \mathbb{E} \cosh(d_{\mathcal{Y}}(\lambda_a, \lambda_b))| \geq C_0 \mathbb{E}|\hat{\mathbb{E}} d_{\mathcal{Y}}^2(\lambda_a, \lambda_b)) - \mathbb{E} d_{\mathcal{Y}}^2(\lambda_a, \lambda_b)|$; that is, $C_0$ reflects the preservation of concentration when moving from distribution $\cosh(d)$ to $d^2$. Then,

**Theorem 3.** *Let $\mathcal{M}$ be a hyperbolic manifold. Fix $0 < \delta < 1$ and let $\Delta(\delta) = \min_{\rho} Pr\left(\forall i, d_{\mathcal{Y}}(\lambda_{a,i}, \lambda_{b,i}) \leq \rho\right) \geq 1 - \delta$. Then, there exists a constant $C_1$ so that with probability at least $1 - \delta$, $\mathbb{E}|\hat{\mathbb{E}} d_{\mathcal{Y}}^2(\lambda_a, y)) - \mathbb{E} d_{\mathcal{Y}}^2(\lambda_a, y)| \leq C_1 \cosh(\Delta(\delta))^{3/2}/C_0 \sqrt{2n}$.*

As we hoped, our estimator is consistent. Note that we pay a price for a tighter bound: $\Delta(\delta)$ is large for smaller probability $\delta$. It is possible to estimate the size of $\Delta(\delta)$ (more generally, it is a function of the curvature). In addition, it is possible to replace the $\Delta(\delta)$ term by applying a version of McDiarmid's inequality for unbounded spaces as in [Kon14].

Next, we adapt the downstream model predictor (2) in the following way. Let $\hat{\mu}_a^2 = \hat{\mathbb{E}}[d_{\mathcal{Y}}^2(\lambda_a, y)]$. Let $\beta = [\beta_1, \ldots, \beta_m]^T$ be such that $\sum_a \beta_a = 1$ and $\beta$ minimizes $\sum_a \beta_a^2 \hat{\mu}_a^2$. Then, we set

$$\tilde{f}(x) = \arg \min_{y \in \mathcal{Y}} \frac{1}{n} \sum_{i=1}^{n} \alpha_i(x) \sum_{a=1}^{m} \beta_a^2 d_{\mathcal{Y}}^2(y, \lambda_{a,i}).$$

We simply replace each of the true labels with a combination of the labeling functions. With this, we can state our final result. First, we introduce our assumptions.

Let $q = \arg \min_{z \in \mathcal{Y}} \mathbb{E}[\alpha(x)(y) d_{\mathcal{Y}}^2(z, y)]$, where the expectation is taken over the population level distribution and $\alpha(x)(y)$ denotes the kernel at $y$.

**Assumption 7.** *(Bounded Hugging Function c.f. [Str20]) Let $q$ be defined as above. For all $a, b \in \mathcal{M}$, the hugging function at $q$ is given by $k_q^b(a) = 1 - (\|\log_q(a) - \log_q(b)\|^2 - d_{\mathcal{Y}}^2(a, b))/d_{\mathcal{Y}}^2(q, b)$. We assume that $k_q^b(a)$ is lower bounded by $k_{\min}$.*

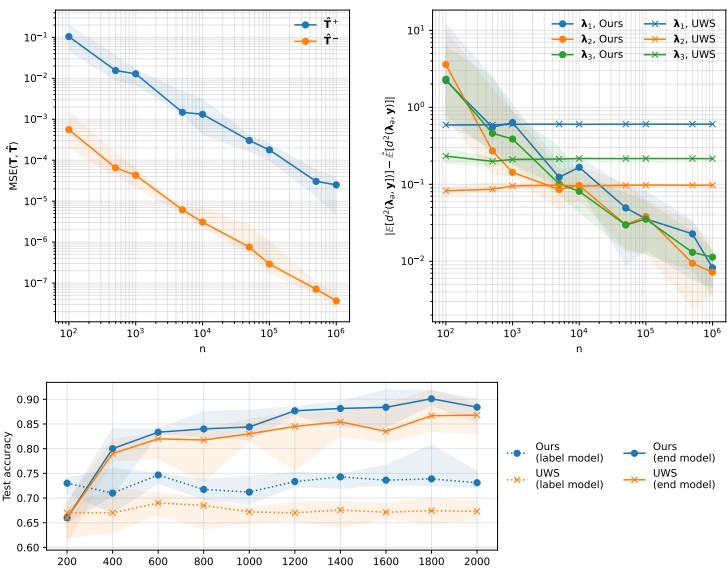

Figure 2: **Finite metric space case.** Parameter estimation improves with samples $n$ in learning to rank—showing nearly-consistent behavior. Our tensor decomposition estimator outperforms [SLV$^+$22]. In particular, (top left) as the number of samples increases, our estimates of the positive and negative components of $\mathbf{T}$ improve. (Top right) the improvements in $\mathbf{T}$ recovery with more samples translates to significantly improved performance over [SLV$^+$22], which is close to constant across $n$. (Bottom) this improved parameter estimation further translates to improvements in label model accuracy (using only the noisy estimates for prediction, without training an end model) and end model generalization. For the top two plots, we use $\boldsymbol{\theta} = [6, 3, 8]$, and in the bottom plot, we use $\boldsymbol{\theta} = [0, 0, 1]$. In all plots, we report medians along with upper and lower quartiles across 10 trials.

**Assumption 8.** *(Kernel Symmetry) We assume that for all $x$ and all $v \in T_q\mathcal{M}$, $\alpha(x)(\exp_q(v)) = \alpha(x)(\exp_q(-v))$.*

The first condition provides control on how geodesic triangles behave; it relates to the curvature. We provide more details on this in the Appendix. The second assumption restricts us to kernels symmetric about the minimizers of the objective $F$. Finally, suppose we draw $(x, y)$ and $(x', y')$ independently from $P_{XY}$. Set $\sigma_o^2 = \alpha(x)(y)\mathbb{E}d_{\mathcal{Y}}^2(y, y')$.

**Theorem 4.** *Let $\mathcal{M}$ be a complete manifold and suppose the assumptions above hold. Then, there exist constants $C_3$, $C_4$ such that,*

$$\mathbb{E}[d_{\mathcal{Y}}^2(\hat{f}(x), \tilde{f}(x))] \leq \frac{C_3\sigma_o^2 + C_4\sum_{a=1}^m \beta_a^2(\hat{\mu}_a^2 + \sigma_o^2)}{n(1 - k_{\min})^2}.$$

Note that as $n$ grows, as long as our worst-quality LF has bounded variance, our estimator of the true predictor is consistent. Moreover, we also have favorable dependence on the noise rate. This is because the only error we incur is in computing suboptimal $\beta$ coefficients. We comment on this suboptimality in the Appendix.

A simple corollary of Theorem 4 provides the generalization guarantees we sought,

**Corollary 1.** *Let $\mathcal{M}$ be a complete manifold and suppose the assumptions above hold. Then, with high probability, $R(\tilde{f}) \leq R(f^*) + \mathcal{O}(n^{-\frac{1}{4}})$.*

**Extensions to Other Manifolds** First, we note that all of our approaches almost immediately lift to products of constant-curvature spaces. For example, we have that $\mathcal{M}_1 \times \mathcal{M}_2$ has metric $d_{\mathcal{Y}}^2(p, q) = d_{\mathcal{M}_1}^2(p_1, q_1) + d_{\mathcal{M}_2}^2(p_2, q_2)$, where $p_i, q_i$ are the projections of $p, q$ onto the $i$th component.

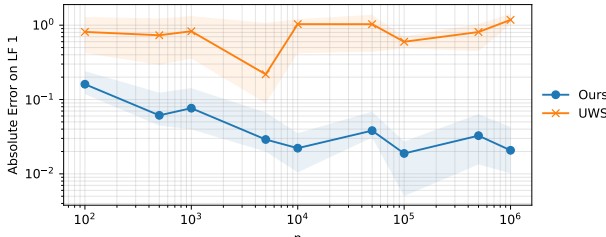

Figure 3: **Continuous case.** Parameter estimation improves with more samples in the hyperbolic regression problem. Our estimator outperforms [SLV$^+$22]. Here, we use different randomly sampled values of $\theta$ for each run. We report medians along with upper and lower quartiles across 10 trials.

We can go beyond products of constant-curvature spaces as well. To do so, we can build generalizations of the law of cosines (as needed for the invariance in Lemma 1). For example, it is possible to do so for symmetric Riemannian manifolds using the tools in [AH91].

## 6 Experiments

Finally, we validate our theoretical claims with experimental results demonstrating improved parameter recovery and end model generalization using our techniques over that of prior work [SLV$^+$22]. We illustrate both the finite metric space and continuous space cases by targeting rankings (i.e., permutations) and hyperbolic spaces. In the case of rankings we show that our pseudo-Euclidean embeddings with tensor decomposition estimator yields stronger parameter recovery and downstream generalization than [SLV$^+$22]. In the case of hyperbolic regression (an example of a Riemannian manifold), we show that our estimator yields improved parameter recovery over [SLV$^+$22].

**Finite metric spaces: Learning to rank**  To experimentally evaluate our tensor decomposition estimator for finite metric spaces, we consider the problem of learning to rank. We construct a synthetic dataset whose ground truth comprises $n$ samples of two distinct rankings among the finite metric space of all length-four permutations. We construct three labeling functions by sampling rankings according to a Mallows model, for which we obtain pseudo-Euclidean embeddings to use with our tensor decomposition estimator.

In Figure 2 (top left), we show that as we increase the number of samples, we can obtain an increasingly accurate estimate of $\mathbf{T}$—exhibiting the *nearly-consistent* behavior predicted by our theoretical claims. This leads to downstream improvements in parameter estimates, which also become more accurate as $n$ increases. In contrast, we find that the estimates of the same parameters given by [SLV$^+$22] do not improve substantially as $n$ increases, and are ultimately worse (see Figure 2, top right). Finally, this leads to improvements in the label model accuracy as compared to that of [SLV$^+$22], and translates to improved accuracy of an end model trained using synthetic samples (see Figure 2, bottom).

**Riemannian manifolds: Hyperbolic regression**  We similarly evaluate our estimator using synthetic labels from a hyperbolic manifold, matching the setting of Section 5. As shown in Figure 3, we find that our estimator consistently outperforms that of [SLV$^+$22], often by an order of magnitude.

## 7 Conclusion

We studied the theoretical properties of weak supervision applied to structured prediction in two general scenarios: label spaces that are finite metric spaces or constant-curvature manifolds. We introduced ways to estimate the noise rates of labeling functions, achieving consistency or near-consistency. Using these tools, we established that with suitable modifications downstream structured prediction models maintain generalization guarantees. Future directions include extending these results to even more general manifolds and removing some of the assumptions needed in our analysis.

## Acknowledgments

We are grateful for the support of the NSF (CCF2106707), the American Family Funding Initiative and the Wisconsin Alumni Research Foundation (WARF). We are thankful to Changho Shin and Harshavardhan Adepu for the discussions and feedback.

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
