# Appendix

The Appendix is organized as follows. First, we provide a glossary that summarizes the notation we use throughout the paper. Afterwards, we provide the proofs for the finite-valued metric space cases. We continue with the proofs and additional discussion for the manifold-valued label spaces. Finally, we give some additional explanations for pseudo-Euclidean spaces.

## A  Glossary

The glossary is given in Table 1 below.

| Symbol | Definition |
|--------|-----------|
| $\mathcal{X}$ | feature space |
| $\mathcal{Y}$ | label metric space |
| $\mathcal{Y}_s$ | support of prior distribution on true labels |
| $d_{\mathcal{Y}}$ | label metric (distance) function |
| $x_1, x_2, \ldots, x_n$ | unlabeled datapoints from $\mathcal{X}$ |
| $y_1, y_2, \ldots, y_n$ | latent (unobserved) labels from $\mathcal{Y}$ |
| $s_1, s_2, \ldots, s_m$ | labeling functions / sources |
| $\lambda_1, \lambda_2, \ldots, \lambda_m$ | output of labeling functions (LFs) |
| $\boldsymbol{\lambda}_1, \boldsymbol{\lambda}_2, \ldots, \boldsymbol{\lambda}_m$ | pseudo-Euclidean embeddings of LFs outputs |
| $\lambda_{a,i}$ | output of $a$th LF on $i$th data point $x_i$ |
| $\boldsymbol{\lambda}_{a,i}$ | pseudo-Euclidean embedding of output of $a$th LF on $i$th data point $x_i$ |
| $n$ | number of data points |
| $m$ | number of LFs |
| $k$ | size of the support of prior on $\mathcal{Y}$ i.e. $k = |S_Y|$ |
| $r$ | size of $\mathcal{Y}$ for the finite case |
| $\theta_a, \hat{\theta}_a$ | true and estimated canonical parameters of model in (3) |
| $\boldsymbol{\theta}, \hat{\boldsymbol{\theta}}$ | true and estimated canonical parameters arranged as vectors |
| $\mathbb{E}[d_{\mathcal{Y}}^2(\lambda_a, y)]$ | mean parameters in (3) |
| $g$ | pseudo-Euclidean embedding mapping |
| $\mathbf{P}$ | true noise model $P_{ij} = P_{\boldsymbol{\theta}}(\tilde{Y} = y_i | Y = y_j)$ with true parameters $\boldsymbol{\theta}$ |
| $\mathbf{Q}$ | estimated noise model with parameters $\hat{\boldsymbol{\theta}}$, $Q_{ij} = P_{\hat{\boldsymbol{\theta}}}(\tilde{Y} = y_i | Y = y_j)$ |
| $\Lambda$ | a random element in $\mathcal{Y}^m$ the $m$-fold Cartesian product of $\mathcal{Y}$ |
| $\Lambda^{(u)}$ | $u$th element in $\mathcal{Y}^m$ |
| $\boldsymbol{\mu}_{a,y}^+, \boldsymbol{\mu}_{a,y}^-$ | means of distributions in (4) corresponding to $\mathbb{R}^{d^+}, \mathbb{R}^{d^-}$ |
| $\epsilon(d^+), \epsilon(d^-)$ | error in recovering the mean parameters (6) |
| $\sigma$ | proxy noise variance in (4) |
| $F(x,y)$ | the score function in (2) with true labels |
| $\tilde{F}_p(x,y), \tilde{F}_q(x,y)$ | the score function in (9) with noisy labels from distributions $\mathbf{P}$ and $\mathbf{Q}$ |
| $\hat{f}$ | minimizer of $F$ defined in (2) |
| $\hat{f}_p, \hat{f}_q$ | minimizers of $\tilde{F}_p, \tilde{F}_q$ as defined in (2) |
| $\sigma_{\max}(\mathbf{P})$ | maximum singular value of $\mathbf{P}$ |
| $\sigma_{\min}(\mathbf{P})$ | minimum singular value of $\mathbf{P}$ |
| $\kappa(\mathbf{P})$ | the condition number of matrix $\mathbf{P}$ |
| $\phi(\mathbf{u}, \mathbf{v})$ | angle between vectors $\mathbf{u}, \mathbf{v} \in \mathbb{R}^d$ |

Table 1: Glossary of variables and symbols used in this paper.

# B Proofs for Parameter Estimation Error in Discrete Spaces

We introduce results leading to the proofs of the theorems for the finite-valued metric space case.

**Lemma 2.** *([AGJ14]) Let $\hat{\mathbf{T}}^+, \hat{\mathbf{T}}^-$ be the third order observed moments for mutually independent labeling functions triplet, as defined in (5) using a sufficiently large number $n$ of i.i.d observations drawn from models in equation (4). Suppose there are sufficiently many such triplets to cover all labeling functions. Let $\hat{\boldsymbol{\mu}}_{a,y}^+, \hat{\boldsymbol{\mu}}_{a,y}^-$ be the estimated parameters returned by the algorithm 1 for all $a \in [m]$. Let $\epsilon(d)$ be defined as above in equation (6), then the following holds with high probability for all labeling functions,*

$$||\boldsymbol{\mu}_{a,y}^+ - \hat{\boldsymbol{\mu}}_{a,y}^+||_2 \leq \mathcal{O}(\epsilon(d^+)) \quad and \quad ||\boldsymbol{\mu}_{a,y}^- - \hat{\boldsymbol{\mu}}_{a,y}^-||_2 \leq \mathcal{O}(\epsilon(d^-)) \quad \forall a \in [m] \; \forall y \in \mathcal{Y}_s \quad (12)$$

*Proof.* The result follows by first showing that our setting and assumptions imply that the conditions of Theorems 1 and 5 in [AGJ14] are satisfied, which allows us to adopt their results. We then translate the result in order to state it in terms of the $\ell_2$ distance.

The tensor concentration result in Theorem 1 in [AGJ14] relies heavily on the noise matrices satisfying the Restricted Isometry Property (RIP) property. The authors make an explicit assumption that the noise model satisfies this condition. In our setting, we have a specific form of the noise model that allows us to show that this assumption is satisfied. The RIP condition is satisfied for sub-Gaussian noise matrices [BDDH14]. Our noise matrices are supported on a discrete space and have bounded entries, and so are sub-Gaussian.

The other required conditions on the norms of factor matrices and the number of latent factors are implied by Assumption 1. Thus, we can adopt the results on recovery of parameters $\boldsymbol{\mu}_{a,y}$ and the prior weights $w_y$ from [AGJ14]. The result gives us for all $a \in [m], y \in \mathcal{Y}_s$,

$$\text{dist}(\boldsymbol{\mu}_{a,y}^+, \hat{\boldsymbol{\mu}}_{a,y}^+) \leq \mathcal{O}(\epsilon(d^+)), \quad \text{dist}(\boldsymbol{\mu}_{a,y}^-, \hat{\boldsymbol{\mu}}_{a,y}^-) \leq \mathcal{O}(\epsilon(d^-)),$$

and

$$|w_y - \hat{w}_y| \leq \mathcal{O}\Big( \max\big(\epsilon(d^+), \epsilon(d^-)\big)/k \Big),$$

where $\text{dist}(\mathbf{u}, \mathbf{v})$ is defined as follows. For any $\mathbf{u}, \mathbf{v} \in \mathbb{R}^d$,

$$\text{dist}(\mathbf{u}, \mathbf{v}) = \sup_{\mathbf{z} \perp \mathbf{u}} \frac{\langle \mathbf{z}, \mathbf{v} \rangle}{||\mathbf{z}||_2 ||\mathbf{v}||_2} = \sup_{\mathbf{z} \perp \mathbf{v}} \frac{\langle \mathbf{z}, \mathbf{u} \rangle}{||\mathbf{z}||_2 ||\mathbf{u}||_2}.$$

Next, we translate the result to the Euclidean distance. For $\mathbf{u}, \mathbf{v} \in \mathbb{R}^d$ with $||\mathbf{u}||, ||\mathbf{v}|| = 1$, it is easy to see that

$$\min_{z \in \{-1, +1\}} ||z\mathbf{u} - \mathbf{v}||_2 \leq \sqrt{2} \, \text{dist}(\mathbf{u}, \mathbf{v}).$$

This notion of distance is oblivious to sign recovery. However, when sign recovery is possible then the Euclidean distance can be bounded as follows,

$$||\mathbf{u} - \mathbf{v}||_2 \leq \sqrt{2} \, \text{dist}(\mathbf{u}, \mathbf{v}).$$

Next we make use of Assumption 2 to recover the signs of $\boldsymbol{\mu}^+, \boldsymbol{\mu}^-$. The assumption bounds the angle between true $\boldsymbol{\mu}_{a,y}^+$ and $\mathbf{y}^+$ between $[0, \pi/2 - c)$ for some sufficiently small $c \in (0, \pi/4]$ such that $\sin(c) > \max(\epsilon_0(d^+), \epsilon_0(d^-))$, where $\epsilon_0(d)$ is defined for some $n_0 < n$ samples in equation (6). We measure $\phi(\hat{\boldsymbol{\mu}}_{a,y}^+, \mathbf{y}^+)$ and $\phi(-\hat{\boldsymbol{\mu}}_{a,y}^+, \mathbf{y}^+)$ and claim that whichever makes an acute angle with $\mathbf{y}^+$ has the correct sign.

We have that $\phi(\hat{\boldsymbol{\mu}}_{a,y}^+, \mathbf{y}^+) \leq \phi(\hat{\boldsymbol{\mu}}_{a,y}^+, \boldsymbol{\mu}_{a,y}^+) + \phi(\boldsymbol{\mu}_{a,y}^+, \mathbf{y}^+)$. Let $s \in \{-1, +1\}$ be the correct sign, then,

$$\begin{aligned}
\phi(s\hat{\boldsymbol{\mu}}_{a,y}^+, \mathbf{y}^+) &\leq \phi(s\hat{\boldsymbol{\mu}}_{a,y}^+, \boldsymbol{\mu}_{a,y}^+) + \phi(s\boldsymbol{\mu}_{a,y}^+, \mathbf{y}^+) \\
&\leq \sin^{-1}(\epsilon(d^+)) + \pi/2 - c \\
&< \pi/2 - (\sin^{-1}(\max(\epsilon_0(d^+), \epsilon_0(d^-))) - \sin^{-1}(\epsilon(d^+))) \\
&< \pi/2 \quad \text{since } \sin^{-1} \text{ is an increasing function in the domain under consideration.}
\end{aligned}$$

With the correct sign $\sin^{-1}(\epsilon(d^+)) < \pi/2$ and so is $\phi(s\hat{\boldsymbol{\mu}}_{a,y}^+, \mathbf{y}^+)$. Thus with incorrect sign $\phi(-s\hat{\boldsymbol{\mu}}_{a,y}^+, \mathbf{y}^+) > \pi/2$.

Hence, after disambiguating the signs we have,

$$||\boldsymbol{\mu}_{a,y}^+ - \hat{\boldsymbol{\mu}}_{a,y}^+||_2 \leq \mathcal{O}(\text{dist}(\boldsymbol{\mu}_{a,y}^+, \boldsymbol{\mu}_{a,y}^-)) \leq \mathcal{O}(\epsilon(d^+))$$

and similarly for $\boldsymbol{\mu}_{a,y}^-$. Next with $n, d$ sufficiently large such that $\epsilon(d^+), \epsilon(d^-) \leq 1$, the result holds for squared distances. $\qquad\square$

**Theorem 1.** *Let $\hat{\boldsymbol{\mu}}_{a,y}^+, \hat{\boldsymbol{\mu}}_{a,y}^-$ be the estimates of $\boldsymbol{\mu}_{a,y}^+, \boldsymbol{\mu}_{a,y}^-$ returned by Algorithm 1 with input $\hat{\mathbf{T}}^+, \hat{\mathbf{T}}^-$ constructed using isometric pseudo-Euclidean embeddings (in $\mathbb{R}^{d^+, d^-}$). Suppose Assumptions 1 and 2 are met, a sufficiently large number of samples $n$ are drawn from the model in (3), and $k = |\mathcal{Y}_s|$. Then there exists a constant $C_0 > 0$ such that with high probability $\forall a \in [m]$ and $y \in \mathcal{Y}_s$,*

$$|\theta_a - \hat{\theta}_a| \leq C_0 \Big| \mathbb{E}[d_{\mathcal{Y}}^2(\lambda_a, y)] - \hat{\mathbb{E}}[d_{\mathcal{Y}}^2(\lambda_a, y)] \Big| \leq \epsilon(d^+) + \epsilon(d^-),$$

*where*

$$\epsilon(d) := \begin{cases} \tilde{\mathcal{O}}\left(k\sqrt{\frac{d}{n}}\right) + \tilde{\mathcal{O}}\left(\frac{\sqrt{k}}{d}\right) & \text{if } \sigma^2 = \Theta(1), \\ \tilde{\mathcal{O}}\left(\sqrt{\frac{k}{n}}\right) + \tilde{\mathcal{O}}\left(\frac{\sqrt{k}}{d}\right) & \text{if } \sigma^2 = \Theta(\frac{1}{d}). \end{cases} \tag{6}$$

*Proof.* We prove this by using the bounds on errors in the estimates of $\boldsymbol{\mu}_{a,y}^+$ and $\boldsymbol{\mu}_{a,y}^-$ from Lemma 2. We proceed by bounding the errors in two parts for $\mathbb{E}[d_\phi^2(\lambda_a^+, \mathbf{y}^+)]$ and $\mathbb{E}[d_\phi^2(\lambda_a^-, \mathbf{y}^-)]$ separately and then combine them to get the bound on overall parameter estimation error.

We first bound the error for $\mathbb{E}[d_\phi^2(\lambda_a^+, \mathbf{y}^+)]$. The true mean parameter (i.e., the true expected squared distance) can be expanded as follows:

$$\mathbb{E}[d_\phi^2(\lambda_a^+, \mathbf{y}^+)] = \mathbb{E}\Big[||\lambda_a^+||_2^2 + ||\mathbf{y}^+||_2^2 - 2\langle \lambda_a^+, \mathbf{y}^+ \rangle\Big],$$
$$= \mathbb{E}_{\boldsymbol{\lambda}}[||\lambda_a^+||_2^2] + \mathbb{E}_{\mathbf{y}}[||\mathbf{y}^+||_2^2] - 2\mathbb{E}_{\mathbf{y}}[\langle \boldsymbol{\mu}_{a,y}^+, \mathbf{y}^+ \rangle].$$

The estimate $\hat{\mathbb{E}}_{\boldsymbol{\lambda}}[||\lambda_a^+||_2^2]$ is computed empirically. The first term is estimated observed LF outputs, i.e. $\hat{\mathbb{E}}_{\boldsymbol{\lambda}}[||\lambda_a^+||_2^2] = \frac{1}{n}\sum_{i=1}^{n} ||\lambda_a^{(i),+}||_2^2$. The second term is computed by using the estimated prior on the labels and for the last term we plug in the estimate of $\boldsymbol{\mu}_{a,y}^+$ computed using the tensor-decomposition algorithm. Putting them all together we have the following estimator:

$$\hat{\mathbb{E}}[d_\phi^2(\lambda_a^+, \mathbf{y}^+)] = \hat{\mathbb{E}}_{\boldsymbol{\lambda}}[||\lambda_a^+||_2^2] + \hat{\mathbb{E}}_{\mathbf{y}}[||\mathbf{y}^+||_2^2] - 2\hat{\mathbb{E}}_{\mathbf{y}}\langle \hat{\boldsymbol{\mu}}_{a,y}^+, \mathbf{y}^+ \rangle.$$

We want to bound the error of our estimator i.e. the difference $|\mathbb{E}[d_\phi^2(\lambda_a^+, \mathbf{y})] - \hat{\mathbb{E}}[d_\phi^2(\lambda_a^+, \mathbf{y})]|$. For this first consider the following,

$$|\mathbb{E}_{\mathbf{y}}\langle \boldsymbol{\mu}_{a,y}^+, \mathbf{y}^+ \rangle - \hat{\mathbb{E}}_{\mathbf{y}}\langle \hat{\boldsymbol{\mu}}_{a,y}^+, \mathbf{y}^+ \rangle| = \sum_y \Big| \langle (w_y \boldsymbol{\mu}_{a,y}^+ - \hat{w}_y \hat{\boldsymbol{\mu}}_{a,y}^+), \mathbf{y}^+ \rangle \Big|$$

$$\leq \sum_y |w_y \langle (\boldsymbol{\mu}_{a,y}^+ - \hat{\boldsymbol{\mu}}_{a,y}^+), \mathbf{y} \rangle| + \sum_y \mathcal{O}(\epsilon(d^+)/k)|\langle \hat{\boldsymbol{\mu}}_{a,y}^+, \mathbf{y} \rangle|$$

$$\leq \sum_y |w_y \langle (\boldsymbol{\mu}_{a,y}^+ - \hat{\boldsymbol{\mu}}_{a,y}^+), \mathbf{y} \rangle| + \mathcal{O}(\epsilon(d^+))$$

$$\leq \sum_y w_y ||\boldsymbol{\mu}_{a,y}^+ - \hat{\boldsymbol{\mu}}_{a,y}^+||_2 ||\mathbf{y}||_2 + \mathcal{O}(\epsilon(d^+))$$

$$\leq \mathcal{O}(\epsilon(d^+)).$$

Here we used $||\boldsymbol{\mu}_{a,y}^+ - \hat{\boldsymbol{\mu}}_{a,y}^+||_2 \leq \mathcal{O}(\epsilon(d^+))$ and $||\boldsymbol{\mu}_{a,y}^+||_2, ||\hat{\boldsymbol{\mu}}_{a,y}^+||_2 = 1, ||\mathbf{y}^+||_2 \leq 1, ||\lambda_a^+||_2^2 \leq 1$ and $|w_y - \hat{w}_y| \leq \mathcal{O}(d^+)/k$. Hence the parameter estimator error,

$$\Big| \mathbb{E}[d_\phi^2(\lambda_a^+, \mathbf{y})] - \hat{\mathbb{E}}[d_\phi^2(\lambda_a^+, \mathbf{y})] \Big| \leq \Big| \mathbb{E}_{\boldsymbol{\lambda}}[||\lambda_a^+||_2^2] - \hat{\mathbb{E}}_{\boldsymbol{\lambda}}[||\lambda_a^+||_2^2] \Big| + 2|\mathbb{E}_{\mathbf{y}}\langle \boldsymbol{\mu}_{a,y}^+, \mathbf{y}^+ \rangle - \hat{\mathbb{E}}_{\mathbf{y}}\langle \hat{\boldsymbol{\mu}}_{a,y}^+, \mathbf{y}^+ \rangle|$$

$$\leq \mathcal{O}(1/\sqrt{n}) + \mathcal{O}(\epsilon(d^+))$$

$$\leq \mathcal{O}(\epsilon(d^+)).$$

In the second step, we bound the first term by $\mathcal{O}(1/\sqrt{n})$ via standard concentration inequalities. Doing the same calculations for $\boldsymbol{\lambda}_a^-$, we obtain

$$\left| \mathbb{E}[d_\phi^2(\boldsymbol{\lambda}_a^-, \mathbf{y})] - \hat{\mathbb{E}}[d_\phi^2(\boldsymbol{\lambda}_a^-, \mathbf{y})] \right| \le \mathcal{O}(\epsilon(d^-)).$$

The overall error in mean parameters is then

$$\left| \mathbb{E}[d_\phi^2(\boldsymbol{\lambda}_a, \mathbf{y})] - \hat{\mathbb{E}}[d_\phi^2(\boldsymbol{\lambda}_a, \mathbf{y})] \right| \le \left| \mathbb{E}[d_\phi^2(\boldsymbol{\lambda}_a^+, \mathbf{y})] - \hat{\mathbb{E}}[d_\phi^2(\boldsymbol{\lambda}_a^+, \mathbf{y})] \right| +$$
$$\left| \mathbb{E}[d_\phi^2(\boldsymbol{\lambda}_a^-, \mathbf{y})] - \hat{\mathbb{E}}[d_\phi^2(\boldsymbol{\lambda}_a^-, \mathbf{y})] \right|,$$
$$\le \mathcal{O}(\epsilon(d^+)) + \mathcal{O}(\epsilon(d^-)).$$

Next, we use a known relation between the mean and the canonical parameters of the exponential model to get the result in terms of the canonical parameters:

$$|\theta_a - \hat{\theta}_a| \le \frac{1}{e_{\min}(A_a(\theta))} \left| \mathbb{E}[d_{\mathcal{Y}}^2(\lambda_a, y)] - \hat{\mathbb{E}}[d_{\mathcal{Y}}^2(\lambda_a, y)] \right|.$$

where $A_a(\theta)$ is the log partition function of the label model in (3) and $e_{\min}(A_a) = \inf_{\theta \in \Theta} \frac{d^2}{d\theta^2} A_a(\theta)$ over the parameter space $\Theta$. For more details see Lemma 8 from [FCS$^+$20] and Theorem 4.3 in [SLV$^+$22]. Letting $C_0 = \max_{a \in [m]} e_{\min}(A_a)$ concludes the proof. $\qquad\square$

## C  Proofs for Generalization Error in Discrete Space

In this section we give the proof for the generalization error bound in the discrete label spaces. We first show that the perturbed (noise-aware) distance function $\tilde{d}_p$ is an unbiased estimator of the true distance. Using this we show that the noise aware score function $\tilde{F}_p$ is a good uniform approximation of the score function $F$. Then we show that the minimizer $\hat{f}_p$ of $\tilde{F}_p$ is close to the minimizer $\hat{f}$ and that this closeness depends on how well $\tilde{F}_p$ approximates $F$. Next, showing that $\tilde{F}_q$ is a good uniform approximation of $\tilde{F}_p$ using the results from previous section on parameter recovery leads to the result on generalization error of $\hat{f}_q$.

**Lemma 3.** *Let the distribution $\tilde{Y}|Y$ be given by $\mathbf{P}$ a $k \times k$ transition probability matrix with $\mathbf{P}_{ij} = \mathbb{P}(\tilde{Y} = y_j | Y = y_i)$ and suppose $\mathbf{P}$ is invertible. Let the pseudo-distance $\tilde{d}_p$ be defined as in* (8) *then,*

$$\mathbb{E}_{\tilde{Y}|Y=y_i}\left[\tilde{d}_p(T, \tilde{Y})\right] = d_{\mathcal{Y}}^2(T, y_i). \tag{13}$$

*Proof.* Set $\tilde{\mathbf{d}}_p \in \mathbb{R}^k$ with $i$th entry $\tilde{\mathbf{d}}_p[i]$ given by $\tilde{d}_p(T, \tilde{Y} = y_i)$ and similarly define $\mathbf{d}$ with $\mathbf{d}[i] = d_{\mathcal{Y}}^2(T, y_i)$. Then we note that $\tilde{\mathbf{d}}_p$ satisfies the following,

$$\tilde{\mathbf{d}}_p = (\mathbf{P})^{-1}\mathbf{d} \implies \mathbb{E}_{\tilde{Y}|Y}[\tilde{\mathbf{d}}_p] = \mathbf{P}(\mathbf{P})^{-1}\mathbf{d} = \mathbf{d},$$

and we are done. $\qquad\square$

Next, we show that the noisy score function $\tilde{F}_p$ concentrates around the true score function $F$ for all $x$ and $y$ with high probability.

**Lemma 4.** *Let $F$ and $\tilde{F}_p$ be defined as in* (9) *and* (2) *over $n$ i.i.d. samples. Then the following holds for any $x \in \mathcal{X}, y \in \mathcal{Y}$ with high probability,*

$$|F(x,y) - \tilde{F}_p(x,y)| \le \tilde{\mathcal{O}}\left(\left(1 + \frac{\sqrt{k}}{\sigma_{\min}(\mathbf{P})}\right)\sqrt{\frac{1}{n}}\right) \quad \forall x \in \mathcal{X}, \forall y \in \mathcal{Y}_s, \tag{14}$$

*where $\sigma_{\min}(\mathbf{P})$ is the minimum singular value of $\mathbf{P}$.*

*Proof.* Let $\{y_i\}_{i=1}^n$ be the true labels of points $\{x_i\}_{i=1}^n$ and let the pseudo-label for $i$th point drawn from the true noise model $\mathbf{P}$ be $\tilde{y}_i$. Recall the definitions of the score functions $F$ and $\tilde{F}_p$ for any $x \in \mathcal{X}$ and $y$ in $\mathcal{Y}$,

$$F(x,y) := \frac{1}{n}\sum_{i=1}^n \alpha_i(x)d_{\mathcal{Y}}^2(y,y_i), \qquad \tilde{F}_p(x,y) := \frac{1}{n}\sum_{i=1}^n \alpha_i(x)\tilde{d}_p(y,\tilde{y}_i).$$

Taking their difference,

$$\tilde{F}_p(x,y) - F(x,y) = \frac{1}{n}\sum_{i=1}^n \alpha_i(x)\Big(\tilde{d}_p(y,\tilde{y}_i) - d_{\mathcal{Y}}^2(y,y_i)\Big),$$

$$= \frac{1}{n}\sum_{i=1}^n \alpha_i(x)\xi(y,y_i,\tilde{y}_i).$$

Here $y, y_i$ are fixed and the randomness is over $\tilde{y}_i$, thus we can think of $\tilde{y}_i$ as random variable $\tilde{Y}_i$ and take the expectation of $\xi$ over the distribution $\mathbf{P}$. From Lemma 3 we have $\mathbb{E}_{\tilde{Y}|Y=y_i}[\xi(y,y_i,\tilde{Y})] = 0$ and this implies $\mathbb{E}[\tilde{F}_p(x,y) - F(x,y)] = 0$.

Moreover, $\alpha_i(x) \cdot \xi(y,y_i,\tilde{Y}_i)$ are independent random variables and $\alpha_i(x) \leq 1$. The $\xi$ are bounded as follows as long as the spectral decomposition of $\mathbf{P}$ is not arbitrary,

$$\max_{z \in \mathcal{Y}_s} \tilde{d}_p(y,z) = ||\tilde{\mathbf{d}}_p||_\infty = ||\mathbf{P}^{-1}\mathbf{d}||_\infty \leq ||\mathbf{P}^{-1}||_\infty ||\mathbf{d}||_\infty.$$

Now using the fact that $||\mathbf{d}||_\infty \leq 1$ and properties of matrix norms we get,

$$||\mathbf{P}^{-1}||_\infty ||\mathbf{d}||_\infty \leq ||\mathbf{P}^{-1}||_\infty \leq \sqrt{k}||\mathbf{P}^{-1}||_2 \leq \frac{\sqrt{k}}{\sigma_{\min}(\mathbf{P})}.$$

Moreover, $\forall y, z \in \mathcal{Y}_s, d_{\mathcal{Y}}^2(y,z) \leq 1$ which gives us the magnitude of random variables $\xi(y,z,\tilde{z})$ is upper bounded by $c_1 := 1 + \frac{\sqrt{k}}{\sigma_{\min}(\mathbf{P})}$ $\forall y, z, \tilde{z} \in \mathcal{Y}_s$. Thus using Hoeffding's inequality and union bound over all $y \in \mathcal{Y}_s$ we get,

$$|\tilde{F}_p(x,y) - F(x,y)| \leq \tilde{\mathcal{O}}\Big(c_1\sqrt{\frac{1}{n}}\Big) \quad \forall y \in \mathcal{Y}_s, x \in \mathcal{X}.$$

Note that, the statement holds for $x \in \mathcal{X}$ without requiring an explicit union bound over $x$. It is because the above concentration depends only on the labels and the events that the above inequality does not hold for any distinct $x_1, x_2 \in \mathcal{X}$ are the same. $\square$

Now, we show that the distance between minimizer of $\tilde{F}_p$ and $F$ is bounded.

**Lemma 5.** *Let $\hat{f}$ be the minimizer as defined in (2) over the clean labels and let $\hat{f}_p$ (defined in eq. (9)) be the minimizer over the noisy labels obtained from conditional distribution $\tilde{Y}|Y$ i.e. $\mathbf{P}$ such that lemma 3, 4 hold, and let the risk function be defined as in (1), then with high probability,*

$$d_{\mathcal{Y}}^2\big(\hat{f}_p(x), \hat{f}(x)\big) \leq \tilde{\mathcal{O}}\Big(\frac{c_1}{\beta}\sqrt{\frac{1}{n}}\Big) \quad \forall x \in \mathcal{X}. \tag{15}$$

*Proof.* Recall the definitions,

$$\hat{f}(x) = \arg\min_{y \in \mathcal{Y}} F(x,y) \qquad \hat{f}_p(x) = \arg\min_{y \in \mathcal{Y}} \tilde{F}_p(x,y)$$

Let $d_{\mathcal{Y}}^2(f_1,f_2) = \sup_{x \in \mathcal{X}} d_{\mathcal{Y}}^2\big(f_1(x), f_2(x)\big)$ and let $\mathcal{B}(\hat{f}, r) = \{f : d_{\mathcal{Y}}^2(\hat{f}, f) \leq r\}$ denote the ball of radius $r$ around $\hat{f}$.

From Lemma 4 we know for $t = \tilde{\mathcal{O}}\left(c_1\sqrt{\frac{1}{n}}\right)$,

$$F\big(x, f(x)\big) - t \leq \tilde{F}_p\big(x, f(x)\big) \leq F\big(x, f(x)\big) + t \quad \forall f : \mathcal{X} \mapsto \mathcal{Y}_s.$$

From Assumption 6 we have,

$$F\big(x, f(x)\big) \geq F\big(x, \hat{f}(x)\big) + \beta \cdot d_{\mathcal{Y}}^2(f(x), \hat{f}(x)).$$

Combining the two we get a lower bound on $\tilde{F}_p$,

$$\tilde{F}_p(x, f(x)) \geq F\big(x, \hat{f}(x)\big) + \beta \cdot d_{\mathcal{Y}}^2(f(x), \hat{f}(x)) - t.$$

We want to find a sufficiently large ball around $\hat{f}$ such that the minimizer of $\tilde{F}_p$ does not lie outside this ball. To see this let $LB$ and $UB$ denote the above mentioned lower and upper bounds on $\tilde{F}_p$,

$$LB(\tilde{F}_p, f, x) := F\big(x, \hat{f}(x)\big) + \beta \cdot d_{\mathcal{Y}}^2(f(x), \hat{f}(x)) - t.$$
$$UB(\tilde{F}_p, f, x) := F\big(x, f(x)\big) + t.$$

For $f \in \mathcal{B}(\hat{f}, \frac{2t}{\beta})$ and some $f'$ such that

$$UB(\tilde{F}_p, f, x) \leq LB(\tilde{F}_p, f', x) \quad \forall x,$$
$$F\big(x, f(x)\big) + t \leq F\big(x, \hat{f}(x)\big) + \beta \cdot d_{\mathcal{Y}}^2(f'(x), \hat{f}(x)) - t,$$
$$F\big(x, f(x)\big) - F\big(x, \hat{f}(x)\big) + t \leq \beta \cdot d_{\mathcal{Y}}^2(f'(x), \hat{f}(x)) - t,$$
$$\beta d_{\mathcal{Y}}^2(f(x), \hat{f}(x)) + t \leq \beta \cdot d_{\mathcal{Y}}^2(f'(x), \hat{f}(x)) - t,$$
$$d_{\mathcal{Y}}^2(f'(x), \hat{f}(x)) \geq 2t/\beta + d_{\mathcal{Y}}^2(f(x), \hat{f}(x)).$$

Thus considering the greatest lower bound, any $f'$ with $d_{\mathcal{Y}}^2(f'(x), \hat{f}(x)) \geq \frac{4t}{\beta}$ cannot be the minimizer of $\tilde{F}_p$, since there exists some other $f$ with smaller distance from $\hat{f}$ that has smaller value compared to $f'$. $\qquad\square$

Next we show that a good estimate of true noise matrix $\mathbf{P}$ by $\mathbf{Q}$ leads to $\tilde{F}_q$ being uniformly close to $\tilde{F}_p$.

**Lemma 6.** *Let* $\mathbf{Q}$, $\mathbf{P}$ *be the distributions defined in equation* (7)*, and* $\tilde{d}_q(T, \tilde{Y})$ *be the distance function as in* (8)*, if* $\max_{ij} |\mathbf{P}_{ij} - \mathbf{Q}_{ij}| = \epsilon$,

$$\left|\tilde{d}_q(y, \tilde{z}_i) - \tilde{d}_p(y, \tilde{z}_i)\right| \leq \mathcal{O}\left(k^2\left(\sigma_{\max}(\mathbf{P}) + \frac{\kappa(\mathbf{P})}{\sigma_{\min}(\mathbf{P})}\right) \cdot \epsilon\right) \qquad \forall y \in \mathcal{Y}_s. \tag{16}$$

*Proof.* Let $\tilde{\mathbf{d}}_q \in \mathbb{R}^k$ be a vector such that its $i^{th}$ entry is given as $\tilde{\mathbf{d}}_q[i] = \tilde{d}_q(T, \tilde{Z} = y_i)$, and similarly, let $\tilde{\mathbf{d}}_p \in \mathbb{R}^k$ with $\tilde{\mathbf{d}}_p[i] = \tilde{d}_p(T, \tilde{Y} = y_i)$, and $\mathbf{d} \in \mathbb{R}^k$ with $\mathbf{d}[i] = d_{\mathcal{Y}}^2(T, Y = y_i)$. It is easy to see that $\tilde{\mathbf{d}}_q = \mathbf{Q}^{-1}\mathbf{d}$ and $\tilde{\mathbf{d}}_p = \mathbf{P}^{-1}\mathbf{d}$. Now consider the following expectation w.r.t $\mathbf{P}$,

$$\tilde{\mathbf{d}}_q - \tilde{\mathbf{d}}_p = \mathbf{Q}^{-1}\mathbf{d} - \mathbf{P}^{-1}\mathbf{d} = \big(\mathbf{Q}^{-1} - \mathbf{P}^{-1}\big)\mathbf{d}.$$

Let $\Delta\mathbf{P} = \mathbf{P} - \mathbf{Q}$, and using standard matrix inversion results for small perturbations, [Dem92], and $||\mathbf{d}||_\infty \leq 1$ we get the following. As $\max_{ij}(\Delta\mathbf{P})_{ij} \leq \epsilon$, we have $||\Delta\mathbf{P}||_2 \leq ||\Delta\mathbf{P}||_F \leq \epsilon k$

$$\begin{aligned}
||\tilde{\mathbf{d}}_p - \tilde{\mathbf{d}}_q||_\infty &\leq ||(\mathbf{P} + \Delta\mathbf{P})^{-1} - \mathbf{P}^{-1}||_\infty ||\mathbf{d}||_\infty, \\
&\leq \sqrt{k}||(\mathbf{P} + \Delta\mathbf{P})^{-1} - \mathbf{P}^{-1}||_2 ||\mathbf{d}||_\infty, \\
&= \sqrt{k}\Big(\kappa(\mathbf{P})||\mathbf{P}^{-1}||_2 ||\Delta\mathbf{P}||_2\Big) + \sqrt{k}\mathcal{O}(||\Delta\mathbf{P}||_2^2), \\
&\leq \sqrt{k} \cdot \kappa(\mathbf{P})||\mathbf{P}^{-1}||_2 \cdot \epsilon k + \mathcal{O}(\epsilon^2 k^{5/2}), \\
&\leq \mathcal{O}\Big(k^{5/2}\Big(1 + \frac{\kappa(\mathbf{P})}{\sigma_{\min}(\mathbf{P})}\Big) \cdot \epsilon\Big) =: c_2.
\end{aligned}$$

$\qquad\square$

**Lemma 7.** *For $\tilde{F}_p$ and $\tilde{F}_q$ defined in (9) w.r.t. noise distributions $\mathbf{P}$ and $\mathbf{Q}$ respectively, and let $\max_{ij} |\mathbf{P}_{ij} - \mathbf{Q}_{ij}| \le \epsilon$ then we have w.h.p.*

$$|\tilde{F}_p(x,y) - \tilde{F}_q(x,y)| \le \tilde{\mathcal{O}}\Big((2c_1 + c_2)\sqrt{\frac{1}{n}}\Big) \qquad \forall y \in \mathcal{Y}_s, \forall x \in \mathcal{X}. \tag{17}$$

*with $c_2 = k^{5/2} \cdot \epsilon \cdot \Big(1 + \frac{\kappa(\mathbf{P})}{\sigma_{\min}(\mathbf{P})}\Big)$ and $c_1 = 1 + \frac{\sqrt{k}}{\sigma_{\min}(\mathbf{P})}$,*

*Proof.* Recall, random variables $\tilde{Y}, \tilde{Z}$ denote the noisy labels drawn from true and estimated noise distributions $\mathbf{P}, \mathbf{Q}$ respectively and $\tilde{y}_i, \tilde{z}_i$ denote their draw for data point $x_i$. Note that we do not know $\mathbf{P}$ and $\tilde{y}_i$ in practice and we only know $\mathbf{Q}, \tilde{z}_i$. Here we are using $\mathbf{P}$ and $\tilde{y}_i$ to compare our actual estimates using samples $\tilde{z}_i$ against the estimates one could have obtained from $\tilde{y}_i$.

Recall the definitions,

$$\tilde{F}_p(x,y) := \frac{1}{n}\sum_{i=1}^{n}\alpha_i(x)\tilde{d}_p(y,\tilde{y}_i), \qquad \tilde{F}_q(x,y) := \frac{1}{n}\sum_{i=1}^{n}\alpha_i(x)\tilde{d}_q(y,\tilde{z}_i).$$

Then,

$$\tilde{F}_p(x,y) - \tilde{F}_q(x,y) = \frac{1}{n}\sum_{i=1}^{n}\alpha_i(x)\Big(\tilde{d}_p(y,\tilde{y}_i) - \tilde{d}_q(y,\tilde{z}_i)\Big) = \frac{1}{n}\sum_{i=1}^{n}\alpha_i(x)\xi(y,\tilde{y}_i,\tilde{z}_i).$$

Thus,

$$\mathbb{E}_{\tilde{Y},\tilde{Z}|Y=y_i}\big[\tilde{d}_p(y,\tilde{Y}) - \tilde{d}_q(y,\tilde{Z})\big] = \mathbb{E}_{\tilde{Z}|Y=y_i}\big[\tilde{d}_q(y,\tilde{Y})\big] - \mathbb{E}_{\tilde{Z}|Y=y_i}\big[\tilde{d}_q(y,\tilde{Z})\big]$$
$$= d_{\mathcal{Y}}^2(y,y_i) - d_{\mathcal{Y}}^2(y,y_i) = 0$$

Finally $\mathbb{E}_{\tilde{Y},\tilde{Z}}[\xi(y,\tilde{Y},\tilde{Z})] = 0$.

Next,

$$\begin{aligned}
\tilde{d}_p(y,\tilde{y}_i) - \tilde{d}_q(y,\tilde{z}_i) &\le |\tilde{d}_p(y,\tilde{y}_i) - \tilde{d}_q(y,\tilde{z}_i)| \\
&\le |\tilde{d}_p(y,\tilde{y}_i) - \tilde{d}_p(y,\tilde{z}_i) + \tilde{d}_p(y,\tilde{z}_i) - \tilde{d}_q(y,\tilde{z}_i)| \\
&\le |\tilde{d}_p(y,\tilde{y}_i) - d_{\mathcal{Y}}^2(y,\tilde{z}_i) + d_{\mathcal{Y}}^2(y,\tilde{z}_i) - \tilde{d}_p(y,\tilde{z}_i) + \tilde{d}_p(y,\tilde{z}_i) - \tilde{d}_q(y,\tilde{z}_i)| \\
&\le |\tilde{d}_p(y,\tilde{y}_i) - d_{\mathcal{Y}}^2(y,\tilde{z}_i)| + |d_{\mathcal{Y}}^2(y,\tilde{z}_i) - \tilde{d}_p(y,\tilde{z}_i)| + |\tilde{d}_p(y,\tilde{z}_i) - \tilde{d}_q(y,\tilde{z}_i)| \\
&\le 2c_1 + |\tilde{d}_p(y,\tilde{z}_i) - \tilde{d}_q(y,\tilde{z}_i)| \\
&\le 2c_1 + c_2.
\end{aligned}$$

The first two terms are upper bounded as in Lemma 4 and the last term is bounded using Lemma 6. Since $\alpha_i(x) \le 1$ and $|\xi(y,\tilde{y}_i,\tilde{z}_i)|$ are upper bounded by $2c_1 + c_2$ as shown above, we have that $|\alpha_i(x) \cdot \xi(y,\tilde{y}_i,\tilde{z}_i)| \le 2c_1 + c_2$. $\qquad\square$

**Lemma 8.** *Let $\hat{f}_p$ be the minimizer as defined in (9) over the noisy labels drawn from $\mathbf{P}$, and let $\hat{f}_q$ (defined in eq. (9)) be the minimizer over the noisy labels obtained from conditional distribution $\mathbf{Q}$. Then with high probability,*

$$d_{\mathcal{Y}}^2\big(\hat{f}_q(x), \hat{f}(x)\big) \le \tilde{\mathcal{O}}\Big(\frac{1}{\beta}(3c_1 + c_2)\sqrt{\frac{1}{n}}\Big) \qquad \forall x \in \mathcal{X}. \tag{18}$$

*Proof.* Let $t_1 = \tilde{\mathcal{O}}\Big(c_1\sqrt{\frac{1}{n}}\Big)$ and $t_2 = \tilde{\mathcal{O}}\Big((2c_1 + c_2)\sqrt{\frac{1}{n}}\Big)$, then combining Lemma 7 and 4 we have,

$$F\big(x, f(x)\big) - t_1 - t_2 \le \tilde{F}_q\big(x, f(x)\big) \le F\big(x, f(x)\big) + t_1 + t_2.$$

Then following same argument as in Lemma 5, we get the result. $\qquad\square$

The following lemmas bound the estimation error between noise matrices $\mathbf{P}$ and $\mathbf{Q}$ using the estimation error in the canonical parameters.

**Lemma 9.** *The posterior distribution function $P_{\boldsymbol{\theta}}(Y = y|\Lambda = \Lambda^u)$ is $(2, \ell_\infty)-$Lipshcitz continuous in $\boldsymbol{\theta}$ for any $y \in \mathcal{Y}$ and $\Lambda^u \in \mathcal{Y}^m$.*

$$|P_{\boldsymbol{\theta}_1}(Y = y|\Lambda = \Lambda^u) - P_{\boldsymbol{\theta}_2}(Y = y|\Lambda = \Lambda^u)| \leq 2||\boldsymbol{\theta}_1 - \boldsymbol{\theta}_2||_\infty \qquad \forall \boldsymbol{\theta}_1, \boldsymbol{\theta}_2 \in \mathbb{R}^m.$$

*Proof.* Recall the definition of the posterior distribution,

$$P_{\boldsymbol{\theta}}(Y = y|\Lambda = \Lambda^u) = \frac{p(Y = y_i)P_{\boldsymbol{\theta}}(\Lambda = \Lambda^u|Y = y_i)}{\sum_{y_j \in \mathcal{Y}} p(Y = y_j)P_{\boldsymbol{\theta}}(\Lambda = \Lambda^u|Y = y_j)}.$$

For convenience let $\mathbf{d}^{(u,i)} \in \mathbb{R}^m$ be such that its $a^{th}$ entry $\mathbf{d}_a^{(u,i)} = d_\mathcal{Y}^2(\Lambda_a^u, y_i)$

$$P_{\boldsymbol{\theta}}(Y = y|\Lambda = \Lambda^u) = \frac{P(Y = y_i) \exp(-\boldsymbol{\theta}^T \mathbf{d}^{(u,i)})}{\sum_{y_j \in \mathcal{Y}} P(Y = y_j) \exp(-\boldsymbol{\theta}^T \mathbf{d}^{(u,j)})}.$$

Let $Z_2(\boldsymbol{\theta}) = \sum_{y_j \in \mathcal{Y}} P(Y = y_j) \exp(-\boldsymbol{\theta}^T \mathbf{d}^{(u,j)})$, then

$$-\nabla_{\boldsymbol{\theta}} \log(Z_2(\boldsymbol{\theta})) = \frac{\sum_{y_j \in \mathcal{Y}} \mathbf{d}^{(u,j)} P(Y = y_j) \exp(-\boldsymbol{\theta}^T \mathbf{d}^{(u,j)})}{Z_2(\boldsymbol{\theta})} = \mathbb{E}_{Y|\Lambda}[\mathbf{d}].$$

Since distances are upper bounded by 1, $||\mathbf{d}||_\infty \leq 1$, so $||\mathbb{E}_{Y|\Lambda}[\mathbf{d}]||_\infty \leq 1$.
Now,

$$\nabla_{\boldsymbol{\theta}} \log \left( P_{\boldsymbol{\theta}}(Y = y|\Lambda = \Lambda^u) \right) = -\mathbf{d}^{(u,i)} - \nabla_{\boldsymbol{\theta}} \log(Z_2(\boldsymbol{\theta})).$$

Thus $||\nabla_{\boldsymbol{\theta}} \log \left( P_{\boldsymbol{\theta}}(Y = y|\Lambda = \Lambda^u) \right)||_\infty \leq 2$.

$$\implies |\log \left( P_{\boldsymbol{\theta}_1}(Y = y|\Lambda = \Lambda^u) \right) - \log \left( P_{\boldsymbol{\theta}_2}(Y = y|\Lambda = \Lambda^u) \right)| \leq 2||\boldsymbol{\theta}_1 - \boldsymbol{\theta}_2||_\infty.$$

Using the fact that for any $t_1, t_2 \in [0, 1]$ $|t_1 - t_2| \leq |\log(t_1) - \log(t_2)|$, gives us the result.

$\square$

**Lemma 10.** *The distribution function $P_{\boldsymbol{\theta}}(\Lambda = \Lambda^u|Y = y)$ is $(2, \ell_\infty)-$Lipshcitz continuous in $\boldsymbol{\theta}$ for any $y \in \mathcal{Y}$ and $\Lambda^u \in \mathcal{Y}^m$.*

$$|P_{\boldsymbol{\theta}_1}(\Lambda = \Lambda^u|Y = y) - P_{\boldsymbol{\theta}_2}(\Lambda = \Lambda^u|Y = y)| \leq 2||\boldsymbol{\theta}_1 - \boldsymbol{\theta}_2||_\infty \qquad \forall \boldsymbol{\theta}_1, \boldsymbol{\theta}_2 \in \mathbb{R}^m.$$

*Proof.* Doing the same steps as in the proof of Lemma 9 gives the result. $\square$

**Lemma 11.** *For the noise distributions $\mathbf{P}, \mathbf{Q}$ in (7) with parameters $\boldsymbol{\theta}, \hat{\boldsymbol{\theta}}$ respectively and $\mathcal{Y}$ restricted only to the elements with non-zero prior probability, $\mathcal{Y}' = \{y \in \mathcal{Y} : P(Y = y) > 0\}$ the following holds,*
$$\max_{ij} |\mathbf{P}_{ij} - \mathbf{Q}_{ij}| \leq 4 \cdot k^m ||\boldsymbol{\theta} - \hat{\boldsymbol{\theta}}||_\infty.$$

*Proof.* It is easy to see that for any two bounded functions $f_1, f_2$ with $|f_1(x)| \leq 1, |f_2(x)| \leq 1$ and Lipschitz continuous with constants $L_1, L_2$, the product of them is also Lipschitz continuous but with constant $L_1 + L_2$. Using this fact along with lemma 9 and lemma 10 gives the result,

$$|\mathbf{P}_{ij} - \mathbf{Q}_{ij}| \leq \sum_{\Lambda^u \in \mathcal{Y}'} |P_{\boldsymbol{\theta}}(y_i|\Lambda^u)P_{\boldsymbol{\theta}}(\Lambda^u|y_j) - P_{\hat{\boldsymbol{\theta}}}(y_i|\Lambda^u)P_{\hat{\boldsymbol{\theta}}}(\Lambda^u|y_j)| \leq 4 \cdot k^m ||\boldsymbol{\theta} - \hat{\boldsymbol{\theta}}||_\infty.$$

$\square$

It is important to note that we are restricting the values of $y$ and $\lambda$ to $\mathcal{Y}'$ which is the set of $y$ with non-zero prior probability and by our assumption it is small.

Finally, we restate and prove our generalization error result:

**Theorem 2.** *(Generalization Error ) Let $\hat{f}$ be the minimizer as defined in* (2) *over the clean labels and let $\hat{f}_q$ (defined in* (9)*) be the minimizer over the noisy labels obtained from inference in Algorithm 1. Suppose Assumptions 4,5,6 hold. Then for $\epsilon_2 = k^{5/2} \cdot \tilde{\mathcal{O}}(\epsilon(d^+) + \epsilon(d^-)) \cdot \left(1 + \frac{\kappa(\mathbf{P})}{\sigma_{\min}(\mathbf{P})}\right)$ and $c_1 = 1 + \frac{\sqrt{k}}{\sigma_{\min}(\mathbf{P})}$, with high probability,*

$$R(\hat{f}_q) \le R(f^*) + \mathcal{O}(n^{-\frac{1}{4}}) + \tilde{\mathcal{O}}\Big(\frac{c_1}{\beta} n^{-\frac{1}{2}}\Big) + \tilde{\mathcal{O}}\Big(\frac{3\epsilon_2}{\beta} n^{-\frac{1}{2}}\Big). \tag{11}$$

*Proof.* Recall the definition of risk function,
$$R(f) = \mathbb{E}_{x,y}\big[d_{\mathcal{Y}}^2\big(f(x), y\big)\big].$$

$$
\begin{aligned}
R(\hat{f}_q) &= \mathbb{E}_{x,y}\big[d_{\mathcal{Y}}^2\big(\hat{f}_q(x), y\big)\big], \\
&\le \mathbb{E}_{x,y}\big[d_{\mathcal{Y}}^2\big(\hat{f}_q(x), \hat{f}(x)\big) + d_{\mathcal{Y}}^2(\hat{f}(x), y) + 2d_{\mathcal{Y}}(\hat{f}_q(x), \hat{f}(x)) \cdot d_{\mathcal{Y}}(\hat{f}(x), y)\big], \\
&= \mathbb{E}_x[d_{\mathcal{Y}}^2\big(\hat{f}_q(x), \hat{f}(x)\big)] + R(\hat{f}) + \tilde{\mathcal{O}}(n^{-1/4}), \\
&\le \tilde{\mathcal{O}}\Big(\frac{1}{\beta}(c_1 + c2)\sqrt{\frac{1}{n}} + \frac{c_2}{\beta}\epsilon\Big) + R(\hat{f}) + \tilde{\mathcal{O}}(n^{-1/4}).
\end{aligned}
$$

Using the result from [CRR16],
$$R(\hat{f}) \le R(f^*) + \mathcal{O}(n^{-1/4}).$$

Combining the two we get
$$R(\hat{f}_q) \le R(f^*) + \tilde{\mathcal{O}}(n^{-1/4}) + \tilde{\mathcal{O}}\Big(\frac{1}{\beta}(c_1 + c2)\sqrt{\frac{1}{n}} + \frac{c_3}{\beta}\epsilon\Big).$$

We get the end result by plugging in the bound on $\epsilon = \max_{ij} ||\mathbf{P} - \mathbf{Q}||$ from Lemma 11 and the bound on parameter recovery error $||\boldsymbol{\theta} - \hat{\boldsymbol{\theta}}||_\infty$ from Theorem 1.

$\square$

## D  Proofs for Continuous Label Spaces

Next we present the proofs for the results in the continuous (manifold-valued) label spaces. We restate the first result on invariance:

**Lemma 1.** *For $\mathcal{Y} = \mathcal{M}$, a hyperbolic manifold, $y \sim P$ for some distribution $P$ on $\mathcal{M}$ and labeling functions $\lambda_a, \lambda_b$ drawn from* (3)*, $\mathbb{E}\cosh d_{\mathcal{Y}}(\lambda_a, \lambda_b) = \mathbb{E}\cosh d_{\mathcal{Y}}(\lambda_b, y)\mathbb{E}\cosh d_{\mathcal{Y}}(\lambda_b, y)$, while for $\mathcal{Y} = \mathcal{M}$ a spherical manifold, $\mathbb{E}\cos d_{\mathcal{Y}}(\lambda_a, \lambda_b) = \mathbb{E}\cos d_{\mathcal{Y}}(\lambda_b, y)\mathbb{E}\cos d_{\mathcal{Y}}(\lambda_b, y)$.*

*Proof.* We start with the hyperbolic law of cosines, which states that
$$\cosh d(\lambda_a, \lambda_b) = \cosh d(\lambda_a, y)\cosh d(\lambda_b, y) + \sinh d(\lambda_a, y)\sinh d(\lambda_b, y)\cos\alpha,$$

where $\alpha$ is the angle between the sides of the triangle formed by $(y, \lambda_a)$ and $(y, \lambda_b)$. We can rewrite this as follows. Let $v_a = \log_y(\lambda_a)$, $v_b = \log_y(\lambda_b)$ be tangent vectors in $T_y M$. Then,
$$\cosh d(\lambda_a, \lambda_b) = \cosh d(\lambda_a, y)\cosh d(\lambda_b, y) + (\sinh\|v_a\|\sinh\|v_b\|)\langle\frac{v_a}{\|v_a\|}, \frac{v_b}{\|v_b\|}\rangle.$$

Next, we take the expectation conditioned on $y$. The right-most term is then
$$
\begin{aligned}
\mathbb{E}[(\sinh\|v_a\|\sinh\|v_b\|)\langle\frac{v_a}{\|v_a\|}, \frac{v_b}{\|v_b\|}\rangle|y] \\
= \mathbb{E}[(\sinh\|v_a\|\sinh\|v_b\|)|y]\mathbb{E}[\langle\frac{v_a}{\|v_a\|}, \frac{v_b}{\|v_b\|}\rangle|y] \\
= 0,
\end{aligned}
$$

where the last equality follows from the fact that $v_a$ and $v_b$ are independent conditioned on $y$ and their distributions are symmetric. This leaves us with the $\cosh$ product terms. Taking expectation again with respect to $y$ gives the result.

The spherical version of the result is nearly identical, replacing hyperbolic sines and cosines with sines and cosines, respectively. $\qquad\square$

Note, in addition, that it is easy to obtain a version of this result for curvatures that are not equal to $-1$ in the hyperbolic case (or $+1$ in the spherical case).

We will use this result for our consistency result, restated below.

**Theorem 3.** *Let $\mathcal{M}$ be a hyperbolic manifold. Fix $0 < \delta < 1$ and let $\Delta(\delta) = \min_\rho Pr\left(\forall i, d_{\mathcal{Y}}(\lambda_{a,i}, \lambda_{b,i}) \leq \rho\right) \geq 1 - \delta$. Then, there exists a constant $C_1$ so that with probability at least $1 - \delta$, $\mathbb{E}|\hat{\mathbb{E}}d_{\mathcal{Y}}^2(\lambda_a, y)) - \mathbb{E}d_{\mathcal{Y}}^2(\lambda_a, y)| \leq C_1 \cosh(\Delta(\delta))^{3/2}/C_0\sqrt{2n}$.*

*Proof.* [Kon14] First, we will condition on the event that the observed outputs have maximal distance (i.e., diameter) $\Delta(\delta)$. This implies that our statements hold with high probability. Then, we use McDiarmid's inequality. For each pair of distinct LFs $a, b$, we have that

$$P\left(\frac{1}{n}|\sum_{i=1}^n \cosh(d(\lambda_{a,i}, \lambda_{b,i})) - \mathbb{E}\cosh(d(\lambda_a, \lambda_b))| \geq t\right) \leq 2\exp\left(-\frac{2nt^2}{\cosh(\Delta(\delta))}\right),$$

Integrating the expression above in $t$, we obtain

$$\mathbb{E}|\hat{\mathbb{E}}\cosh(d(\lambda_a, \lambda_b)) - \mathbb{E}\cosh(d(\lambda_a, \lambda_b))| \leq \frac{\sqrt{\pi\cosh(\Delta(\delta))}}{\sqrt{2n}}. \tag{19}$$

Next, we use this to control the gap on our estimator. Recall that using the triplet approach, we estimate

$$\hat{\mathbb{E}}\cosh(d(\lambda_a, y)) = \sqrt{\frac{\hat{\mathbb{E}}\cosh d(\lambda_a, \lambda_b)\hat{\mathbb{E}}\cosh d(\lambda_a, \lambda_c)}{(\hat{\mathbb{E}}\cosh d(\lambda_b, \lambda_c))^2}}.$$

For notational convenience, we write $\nu(a)$ for $\mathbb{E}(\cosh(d(\lambda_a, y)))$, $\hat{\nu}(a)$ for its empirical counterpart, and $\nu(a, b)$ and $\hat{\nu}(a, b)$ for the versions between pairs of LFs $a, b$. Then, the above becomes

$$\hat{\nu}(a) = \sqrt{\frac{\hat{\nu}(a, b)\hat{\nu}(a, c)}{(\hat{\nu}(b, c))^2}}.$$

Note that $\cosh(x) \geq 1$, so that $\hat{\nu}(a, b) \geq 1$ and similarly for the empirical versions. We also have that $\hat{\nu}(a, b) \leq \cosh(\Delta(\delta))$. With this, we can begin our perturbation analysis. Applying Lemma 1, we have that

$$\mathbb{E}|\hat{\nu}(a) - \nu(a)| = \mathbb{E}\left|\sqrt{\frac{\hat{\nu}(a, b)\hat{\nu}(a, c)}{\hat{\nu}(b, c)^2}} - \sqrt{\frac{\nu(a, b)\nu(a, c)}{\nu(b, c)^2}}\right|$$

$$= \mathbb{E}\left|\sqrt{\frac{\hat{\nu}(a, b)\hat{\nu}(a, c)}{\hat{\nu}(b, c)^2}} - \sqrt{\frac{\nu(a, b)\hat{\nu}(a, c)}{\hat{\nu}(b, c)^2}} + \sqrt{\frac{\nu(a, b)\hat{\nu}(a, c)}{\hat{\nu}(b, c)^2}} - \sqrt{\frac{\nu(a, b)\nu(a, c)}{\nu(b, c)^2}}\right|$$

$$\leq \mathbb{E}\left|\sqrt{\frac{\hat{\nu}(a, b)\hat{\nu}(a, c)}{\hat{\nu}(b, c)^2}} - \sqrt{\frac{\nu(a, b)\hat{\nu}(a, c)}{\hat{\nu}(b, c)^2}}\right| + \mathbb{E}\left|\sqrt{\frac{\nu(a, b)\hat{\nu}(a, c)}{\hat{\nu}(b, c)^2}} - \sqrt{\frac{\nu(a, b)\nu(a, c)}{\nu(b, c)^2}}\right|$$

$$= \mathbb{E}\left|\sqrt{\frac{\hat{\nu}(a, c)}{\hat{\nu}(b, c)^2}}(\sqrt{\hat{\nu}(a, b)} - \sqrt{\nu(a, b)})\right| + \mathbb{E}\left|\sqrt{\frac{\nu(a, b)\hat{\nu}(a, c)}{\hat{\nu}(b, c)^2}} - \sqrt{\frac{\nu(a, b)\nu(a, c)}{\nu(b, c)^2}}\right|$$

$$\leq \frac{\sqrt{\pi}\cosh(\Delta(\delta))}{\sqrt{2n}} + \mathbb{E}\left|\sqrt{\frac{\nu(a, b)\hat{\nu}(a, c)}{\hat{\nu}(b, c)^2}} - \sqrt{\frac{\nu(a, b)\nu(a, c)}{\nu(b, c)^2}}\right|.$$

To see why the last step holds, note that $\sqrt{\hat{\nu}(a,c)} \leq \sqrt{\cosh(\Delta(\delta))}$, while $\hat{\nu}(b,c) \geq 1$. Next, for $\alpha, \beta \geq 1$, $|\sqrt{\alpha} - \sqrt{\beta}| = \frac{|\alpha-\beta|}{\sqrt{\alpha}+\sqrt{\beta}} \leq |\alpha - \beta|$. This means that $\mathbb{E}|\sqrt{\hat{\nu}(a,b)} - \sqrt{\nu(a,b)}| \leq \mathbb{E}|\hat{\nu}(a,b) - \nu(a,b)| \leq \frac{\sqrt{\pi \cosh(\Delta(\delta))}}{\sqrt{2n}}$ using (19).

Now we can continue, adding and subtracting as before. We have that

$$\mathbb{E}\left|\sqrt{\frac{\nu(a,b)\hat{\nu}(a,c)}{\hat{\nu}(b,c)^2}} - \sqrt{\frac{\nu(a,b)\nu(a,c)}{\nu(b,c)^2}}\right|$$

$$\leq \mathbb{E}\left|\sqrt{\frac{\nu(a,b)\hat{\nu}(a,c)}{\hat{\nu}(b,c)^2}} - \sqrt{\frac{\nu(a,b)\nu(a,c)}{\hat{\nu}(b,c)^2}}\right| + \mathbb{E}\left|\sqrt{\frac{\nu(a,b)\nu(a,c)}{\hat{\nu}(b,c)^2}} - \sqrt{\frac{\nu(a,b)\nu(a,c)}{\nu(b,c)^2}}\right|$$

$$\leq \frac{\sqrt{\pi}\cosh(\Delta(\delta))}{\sqrt{2n}} + \mathbb{E}\left|\sqrt{\frac{\nu(a,b)\nu(a,c)}{\hat{\nu}(b,c)^2}} - \sqrt{\frac{\nu(a,b)\nu(a,c)}{\nu(b,c)^2}}\right|$$

$$\leq \frac{\sqrt{\pi}\cosh(\Delta(\delta))}{\sqrt{2n}} + \frac{2\sqrt{\pi}(\cosh(\Delta(\delta)))^{3/2}}{\sqrt{n}}.$$

The first expectation in the r.h.s is bounded using the same steps as above. The second expectation is bounded as follows,

$$\mathbb{E}\left|\sqrt{\frac{\nu(a,b)\nu(a,c)}{\hat{\nu}(b,c)^2}} - \sqrt{\frac{\nu(a,b)\nu(a,c)}{\nu(b,c)^2}}\right| \leq \mathbb{E}\left|\sqrt{\nu(a,b)\nu(a,c)}\left(\frac{(\hat{\nu}(b,c) - \nu(b,c))(\hat{\nu}(b,c) + \nu(b,c))}{\hat{\nu}(b,c)\nu(b,c)}\right)\right|$$

Here, the denominator is lower bounded by 1 and in the numerator $\sqrt{\nu(a,b)\nu(a,c)} \leq \cosh(\Delta(\delta))$ and $\hat{\nu}(b,c) + \nu(b,c) \leq 2\cosh(\Delta(\delta))$ and $\mathbb{E}(\hat{\nu}(b,c) - \nu(b,c)) \leq \frac{\sqrt{\pi \cosh(\Delta(\delta))}}{\sqrt{2n}}$. Putting it all together, with probability at least $1 - \delta$,

$$\mathbb{E}|\hat{\mathbb{E}}\cosh(d(\lambda_a, y)) - \mathbb{E}\cosh(d(\lambda_a, y))| \leq \frac{2\sqrt{\pi}\cosh(\Delta(\delta)) + 2\sqrt{\pi}(\cosh(\Delta(\delta))^{3/2}}{\sqrt{n}}. \quad (20)$$

Next, recall that $C_0$ satisfies $\mathbb{E}|\hat{\mathbb{E}}\cosh(d(\lambda_a, \lambda_b)) - \mathbb{E}\cosh(d(\lambda_a, \lambda_b))| \geq C_0\mathbb{E}|\hat{\mathbb{E}}d(\lambda_a, \lambda_b)) - \mathbb{E}d(\lambda_a, \lambda_b)|$. Thus,

$$\mathbb{E}|\hat{\mathbb{E}}d^2(\lambda_a, y) - \mathbb{E}d^2(\lambda_a, y)| \leq \frac{2\sqrt{\pi}\cosh(\Delta(\delta)) + 2\sqrt{\pi}(\cosh(\Delta(\delta)))^{3/2}}{C_0\sqrt{n}}.$$

This concludes the proof. □

Next, we will prove a simple result that is needed in the proof of Theorem 5. Consider the distribution $P$ of the quantities $\alpha(x)(y)d_{\mathcal{Y}}^2(z, y)$ for some fixed $z \in \mathcal{M}$. We can think of this as the population-level version of sample distances that are observed in the supervised version of the problem. We do not have access to it in our approach; it will be used only as an object in our proof. Recall we set $q = \arg\min_{z \in \mathcal{Y}} \mathbb{E}[\alpha(x)(y)d_{\mathcal{Y}}^2(z, y)]$ to be the population-level minimizer. Here we use the notation $\alpha(x)(y)$ to denote the corresponding kernel value at a point $y$. Finally, let us denote $P'$ to be the distribution over the quantities $\alpha(x)(y)\sum_{a=1}^{m}\beta_a^2 d_{\mathcal{Y}}^2(z, \lambda_{a,i})$.

**Lemma 12.** *Let the distributions $P$ and $P'$ be defined as above, with $q$ the minimizer of $\mathbb{E}_P[\alpha(x)(y)d_{\mathcal{Y}}^2(z, y)]$. Suppose that Assumptions 7 and 8 hold. Then, $q$ is also the minimizer of $\mathbb{E}_{P'}[\alpha(x)(y)\sum_{a=1}^{m}\beta_a^2 d_{\mathcal{Y}}^2(z, \lambda_{a,i})]$.*

*Proof.* We will use a simple symmetry argument. First, note that the minimizer of the objective function under $P'$ is not affected by uniformly scaling the distances by some constant. If we do so repeatedly, we can shrink the region in which this minimizer—and that of the objective function for $P$—are found. This means that the distance between the two minimizers must be arbitrarily small, so that by a limit argument, they must be the same.

□

Finally, this enables us to prove our main result, Theorem 5, restated below:

**Theorem 5.** *Let $\mathcal{M}$ be a complete manifold and suppose the assumptions above hold. Then, there exist constants $C_3$, $C_4$ such that,*

$$\mathbb{E}[d_{\mathcal{Y}}^2(\hat{f}(x), \tilde{f}(x))] \leq \frac{C_3\sigma_o^2 + C_4\sum_{a=1}^m \beta_a^2(\hat{\mu}_a^2 + \sigma_o^2)}{n(1 - k_{\min})^2}.$$

*Proof.* We use Lemma 12 and compute a bound on the expected distance from the empirical estimates to the common center. In both cases, the approach is nearly identical to that of [Str20] (proof of Theorem 3.2.1); we include these steps for clarity. Suppose that the minimum and maximum values of $\alpha$ are $\alpha_{\min}$ and $\alpha_{\max}$, respectively.

Using the hugging function assumption, we have that,

$$\|\log_q(\hat{f}(x)) - \log_q(y_i)\|^2 \leq k_{\min}d_{\mathcal{Y}}^2(q, \hat{f}(x)) + d_{\mathcal{Y}}^2(\hat{f}(x), y_i).$$

We also have that

$$\|\log_q(\hat{f}(x)) - \log_q(y_i)\|^2 = d_{\mathcal{Y}}^2(q, \hat{f}(x)) - 2\langle\log_q(\hat{f}(x)), \log_q(y_i)\rangle + d_{\mathcal{Y}}^2(q, y_i).$$

Then,

$$(1 - k_{\min})d_{\mathcal{Y}}^2(q, \hat{f}(x)) \leq 2\langle\log_q(\hat{f}(x)), \log_q(y_i)\rangle + d_{\mathcal{Y}}^2(\hat{f}(x), y_i) - d_{\mathcal{Y}}^2(q, y_i).$$

Now, multiply each of the equations by $\alpha_i$ and sum over them. In that case, the difference on the right side is non-positive, as $\hat{f}(x)$ is the empirical minimizer. This yields

$$\sum_{i=1}^n \alpha(x)_i(1 - k_{\min})d_{\mathcal{Y}}^2(q, \hat{f}(x)) \leq \sum_{i=1}^n \alpha(x)_i 2\langle\log_q(\hat{f}(x)), \log_q(y_i)\rangle.$$

Using the minimum and maximum values of $\alpha$, and setting $\bar{q} = \frac{1}{n}\sum_{i=1}^n \log_q(y_i)$, we get

$$\alpha_{\min}(1 - k_{\min})d_{\mathcal{Y}}^2(q, \hat{f}(x)) \leq 2\alpha_{\max}\langle\log_q(\hat{f}(x)), \bar{q}\rangle.$$

We apply Cauchy-Schwarz, obtaining

$$\alpha_{\min}(1 - k_{\min})d_{\mathcal{Y}}^2(q, \hat{f}(x)) \leq 2\alpha_{\max}\|\log_q(\hat{f}(x)\|\|\bar{q}\|.$$

Since $\|\log_q(\hat{f}(x)\| = d_{\mathcal{Y}}(q, \hat{f}(x))$, we then have that

$$\alpha_{\min}(1 - k_{\min})d_{\mathcal{Y}}(q, \hat{f}(x)) \leq 2\alpha_{\max}\|\bar{q}\|.$$

Squaring both sides, we obtain

$$\alpha_{\min}^2(1 - k_{\min})^2 d_{\mathcal{Y}}^2(q, \hat{f}(x)) \leq 4\alpha_{\max}^2\|\bar{q}\|^2.$$

What remains is to take expectation and use the fact that the tangent vectors whose average forms $\bar{q}$ are independent. This yields

$$\alpha_{\min}^2(1 - k_{\min})^2 \mathbb{E}d_{\mathcal{Y}}^2(q, \hat{f}(x)) \leq 4\alpha_{\max}^2\frac{\sigma_o^2}{n}.$$

Thus we obtain

$$\alpha_{\min}^2(1 - k_{\min})^2 \mathbb{E}d_{\mathcal{Y}}^2(q, \hat{f}(x)) \leq 4\alpha_{\max}^2\frac{\sigma_o^2}{n},$$

or

$$\mathbb{E}d_{\mathcal{Y}}^2(q, \hat{f}(x)) \leq 4\frac{\alpha_{\max}^2}{\alpha_{\min}^2}\frac{\sigma_o^2}{n(1 - k_{\min})^2}. \tag{21}$$

We use the same approach, but apply it to the objective function that involves the $n$ samples of the $m$ LFs drawn from the distribution $P'$. In this case, the $\bar{q}$ vector becomes $\frac{1}{n}\sum_{i=1}^n (\sum_{a=1}^m \beta_a \log_q(\lambda_{a,i}))$.

Doing so yields

$$\alpha_{\min}^2 (1 - k_{\min})^2 \mathbb{E} d_{\mathcal{Y}}^2(q, \tilde{f}(x)) \leq 4\alpha_{\max}^2 \frac{\sum_{a=1}^m \beta_a^2 \sigma_a^2}{n},$$

where $\sigma_a^2$ corresponds to the expected squared distance for LF $a$ to $q$. We bound this with triangle inequality, obtaining $\sigma_a^2 \leq 2\sigma_o^2 + 2\hat{\mu}_a^2$, so that

$$\alpha_{\min}^2 (1 - k_{\min})^2 \mathbb{E} d_{\mathcal{Y}}^2(q, \tilde{f}(x)) \leq 8\alpha_{\max}^2 \frac{\sum_{a=1}^m \beta_a^2 (\sigma_o + \hat{\mu}_a^2)}{n},$$

or,

$$\mathbb{E} d_{\mathcal{Y}}^2(q, \tilde{f}(x)) \leq 8 \frac{\alpha_{\max}^2}{\alpha_{\min}^2} \frac{\sum_{a=1}^m \beta_a^2 (\sigma_o^2 + \hat{\mu}_a^2)}{n(1 - k_{\min})^2}. \tag{22}$$

Now, again using triangle inequality,

$$\mathbb{E} d_{\mathcal{Y}}^2(\hat{f}(x), \tilde{f}(x)) \leq 2\mathbb{E} d_{\mathcal{Y}}^2(q, \hat{f}(x)) + 2\mathbb{E} d_{\mathcal{Y}}^2(q, \tilde{f}(x)).$$

Plugging (22) and (21) into this bound produces the result. $\qquad\square$

# E  Additional Details on Continuous Label Space

We provide some additional details on the continuous (manifold-valued) case.

**Computing $\Delta(\delta)$**   In Theorem 3, we stated the result in terms of $\Delta(\delta)$, a quantity that trades off the probability of failure $\delta$ for the diameter of the largest ball that contains the observed points. Note that if we fix the curvature of the manifold, it is possible to compute an exact bound for this quantity by using formulas for the sizes of balls in $d$-dimensional manifolds of fixed curvature.

**Hugging function**   Note that it is possible to derive a lower bound on the hugging function as a function of the curvature. The way to do so is to use *comparison theorems* that upper bound triangle edge lengths with those of larger-curvature triangles. This makes it possible to establish a concrete value for $k_{\min}$ as a function of the curvature.

We note, as well, that an upper bound $k_{\max}$ on the hugging function can be obtained by a simple rearrangement of Lemma 6 from [ZS16]. This result follows from a curvature lower bound based on hyperbolic law of cosines; the bound we describe follows from the opposite—an upper bound based on spherical triangles.

$\beta$ **Weights and Suboptimality**   An intuitive way to think of the estimator we described is the following simple Euclidean version. Suppose we have labeling functions $\lambda_1, \ldots, \lambda_m$ that are equal to $y + \varepsilon_a$, where $\varepsilon_a \sim \mathcal{N}(0, \sigma_a^2)$. In this case, if we seek an unbiased estimator with lowest variance, we require a set of weights $\beta_a$ so that $\sum_a \beta_a = 1$ and $\mathrm{Var}[\frac{1}{m} \sum_{a=1}^m \beta_a \lambda_a]$ is minimized. It is not hard to derive a closed-form solution for the $\beta_a$ coefficients as a function of the terms $\sigma_a^2$.

Now, suppose we use the same solution, but with noisy estimates $\hat{\sigma}^2$ instead. Our weights $\hat{\beta}$ will yield a suboptimal variance, but this will not affect the scaling of the rate in terms of the number of samples $n$.

# F  Extended Background on Pseudo-Euclidean Embeddings

We provide some additional background on pseudo-metric spaces and pseudo-Euclidean embeddings. Our roadmap is as follows. First, we note that pseudo-Euclidean spaces are a particular kind of pseudo-metric space, so we provide additional background and formal definitions for these pseudo-metric spaces. Afterwards, we explain some of the ideas behind pseudo-Euclidean spaces, comparing them to standard Euclidean spaces in the context of embeddings.

### F.1 Pseudo-metric Spaces

Pseudo-metric spaces generalize metric spaces by removing the requirement that pairs of points at distance zero must be identical:

**Definition 1.** (*Pseudo-metric Space*) *A set $\mathcal{Y}$ along with a distance function $d_{\mathcal{Y}} : \mathcal{Y} \times \mathcal{Y} \mapsto \mathbb{R}^+$ is called pseudo-metric space if $d_{\mathcal{Y}}$ satisfies the following conditions,*

$$\forall \mathbf{y}, \mathbf{z} \in \mathcal{Y} \qquad d_{\mathcal{Y}}(\mathbf{y}, \mathbf{z}) = d_{\mathcal{Y}}(\mathbf{y}, \mathbf{z}) \tag{23}$$
(Symmetry)

$$\forall \mathbf{y} \in \mathcal{Y} \qquad d_{\mathcal{Y}}(\mathbf{y}, \mathbf{y}) = 0 \tag{24}$$
(Reflexivity)

$$\forall \mathbf{x}, \mathbf{y}, \mathbf{z} \in \mathcal{Y} \qquad d_{\mathcal{Y}}(\mathbf{y}, \mathbf{x}) \leq d_{\mathcal{Y}}(\mathbf{y}, \mathbf{z}) + d_{\mathcal{Y}}(\mathbf{x}, \mathbf{z}) \tag{25}$$
(Triangle Inequality)

*These spaces have additional flexibility compared to standard metric spaces: note that while $d(y, y) = 0$, $d(x, y) = 0$ does not imply that $x$ and $y$ are identical. The downside of using such spaces, however, is that conventional algebra may not produce the usual results. For example, limits where the distance between a sequence of points and a particular point tends to zero do not convey the same information as in standard metric spaces. However, these odd properties do not concern us, as we only use the spaces for representing a set of distances from our given metric space.*

A finite pseudo-metric space has $|\mathcal{Y}| < \infty$.

### F.2 Pseudo-Euclidean Spaces

The following definitions are for *finite-dimensional* vector spaces defined over the field $\mathbb{R}$.

**Definition 2.** (*Symmetric Bilinear Form / Generalized Inner Product*) *For a vector space $\mathcal{Y}$ over the field $\mathbb{R}$, a symmetric bilinear form is a function $\phi : \mathcal{Y} \times \mathcal{Y} \mapsto \mathbb{R}$ satisfying the following properties $\forall y_1, y_2, z, y \in \mathcal{Y}, c \in \mathbb{R}$:*

*P1)* $\phi(y_1 + y_2, y) = \phi(y_1, y) + \phi(y_2, y)$,

*P2)* $\phi(cy, z) = c\phi(y, z)$,

*P3)* $\phi(y, z) = \phi(z, y)$.

**Definition 3.** (*Squared Distance w.r.t. $\phi$*) *Let $V$ be a real vector space equipped with generalized inner product $\phi$, then the squared distance w.r.t. $\phi$ between any two vectors $\mathbf{y}, \mathbf{z} \in V$ is defined as,*

$$||\mathbf{y} - \mathbf{z}||_\phi^2 := \phi(\mathbf{y} - \mathbf{z}, \mathbf{y} - \mathbf{z})$$

This definition also gives a notion of squared length for every $\mathbf{y} \in V$,

$$||\mathbf{y}||_\phi^2 := \phi(\mathbf{y}, \mathbf{y})$$

The inner product can also be expressed in terms of a basis of the vector space $V$. Let the dimension of $\mathcal{Y}$ be $d$, and $\{\mathbf{b}_i\}_{i=1}^d$ be a basis of $\mathcal{Y}$, then for any two vectors $\mathbf{y} = [y_1, \ldots y_d], \mathbf{z} = [z_1, \ldots z_d] \in V$,

$$\phi(\mathbf{y}, \mathbf{z}) = \sum_{i=1}^{d} \sum_{j=1}^{d} y_i z_i \phi(\mathbf{b}_i, \mathbf{b}_j)$$

The matrix $\mathbf{M}(\phi) := [\phi(\mathbf{b}_i, \mathbf{b}_j)]_{1 \leq i,j \leq d}$ is called *the matrix of* $\phi$ w.r.t the basis $\{\mathbf{b}_i\}_{i=1}^d$ It gives a convenient way to express the inner product as $\phi(\mathbf{y}, \mathbf{z}) = \mathbf{y}^T \mathbf{M}(\phi)\mathbf{z}$. A symmetric bilinear form $\phi$ on a vector space of dimension $d$, is said to be *non-degenerate* if the rank of $\mathbf{M}(\phi)$ w.r.t to some basis is equal to $d$.

Example: For the $d-$ dimensional euclidean space with standard basis and $\phi$ as dot product we get $\mathbf{M}(\phi) = \mathbf{I}_d$

**Definition 4.** (*Pseudo-euclidean Spaces*) *A real vector space $\mathbb{R}^{d^+, d^-}$ of dimension $d = d^+ + d^-$, equipped with a non-degenerate symmetric bilinear form $\phi$ is called a pseudo-euclidean (or Minkowski) vector space of signature $(d^+, d^-)$ if the matrix of $\phi$ w.r.t a basis $\{\mathbf{b}_i\}_{i=1}^d$ of $\mathbb{R}^{d^+, d^-}$, is given as,*

$$\mathbf{M}(\phi) = \begin{pmatrix} \mathbf{I}_{d^+} & \mathbf{0} \\ \mathbf{0} & -\mathbf{I}_{d^-} \end{pmatrix}_{d \times d}$$

**Embedding Algorithms**    The tool that ensures we can produce isometric embeddings is the following result:

**Proposition 1.** *([Gol85]) Let $\mathcal{Y} = \{y_0, \ldots y_k\}$ be a finite pseudo-metric space equipped with distance function $d_{\mathcal{Y}}$, and let $\mathbf{V} = \{\mathbf{v}_i, \ldots, \mathbf{v}_k\}$ be a collection of vectors in $\mathbb{R}^{d^+, d^-}$. Then $\mathcal{Y}$ is isometrically embeddable in $\mathbb{R}^{d^+, d^-}$ if and only if,*

$$\langle \mathbf{v}_i, \mathbf{v}_j \rangle_\phi = \frac{1}{2} \Big( d^2_{\mathcal{Y}}(y_i, y_0) + d^2_{\mathcal{Y}}(y_j, y_0) - d^2_{\mathcal{Y}}(y_i, y_j) \Big) \quad \forall i, j \in [k] \tag{26}$$

This bilinear form is very similar to the one used for MDS embeddings [KW78]—it is closely related to the squared distance matrix. The main information needed is what the signature (i.e., how many positive, negative, and zero eigenvalues) of this bilinear form is. If the dimension of the pseudo-Euclidean space we choose to embed in is at least as large as the number of positive and negative eigenvalues, we can obtain isometric embeddings. Because we are working with finite metric spaces, this number is always finite, and, in fact, is never larger than the size of the metric space. This means we can always produce isometric embeddings.

The practical aspects of how to produce the embedding are shown in the first half of Algorithm 1. The basic idea is to do an eigendecomposition and capture eigenvectors corresponding to the positive and negative eigenvalues. These allow us to perfectly reproduce the positive and negative components of the distances separately; the resulting distance is the difference between the two components. The process of performing the eigendecomposition is standard, so that the overall procedure has the same complexity as running MDS. Compare this to MDS: there, we only capture the eigenvectors corresponding to the positive eigenvalues and ignore the negative ones. Otherwise the procedure is identical.

We note that in fact it is possible to embed pseudo-metric spaces isometrically into pseudo-Euclidean spaces, but we never use this fact. Our only application of this tool is to embed conventional metric spaces. However, our results directly lift to this more general setting.

The idea of using pseudo-Euclidean spaces for embeddings that can then be used in kernel-based or other classifiers or other approaches to machine learning is not new. For example, [PPD01] used these spaces for kernel-based learning, [LRBM06] used them for generic pairwise learning, and [PHD+06] showed that they are among several non-standard spaces that provide high-quality representations. Our contribution is using these in the context of weak supervision and learning latent variable models.

**Dimensionality**    We also give more detail on the example we provided showing that pseudo-Euclidean embeddings can have arbitrarily better dimensionality compared to one-hot encodings. The idea here is simple. We start with a particular kind of tree with a root and three branches that are simply long chains (paths) and have $t$ nodes each, for a total of $3t + 1$ nodes. One-hot encodings have dimension that scales with the number of nodes, i.e., dimension $3t + 1$.

Pseudo-euclidean embeddings enable us to embed such a tree into a space of finite (and in fact, very small) dimension while preserving the shortest-hops distances between each pair of nodes in the graph. As described above, the key question is what the number of positive and negative eigenvalues for the squared distance matrix (and thus the bilinear form) is. Fortunately, for such graphs, the signature of the squared-distance matrix is known (Theorem 20 in [BS16]). Applying this result shows that the pseudo-Euclidean dimension is just 3, a tiny fixed value regardless of the value of $t$ above.