# OpenReview forum: "Lifting Weak Supervision To Structured Prediction"
_NeurIPS.cc/2022/Conference — NeurIPS 2022 Accept_

### Official Review · Reviewer_x4kV · 2022-06-24

**Rating:** 7
**Confidence:** 3
**Soundness:** 3 good
**Presentation:** 3 good
**Contribution:** 3 good

**Summary:**

This paper considers the problem of aggregating noisy labels for learning with structured label set. It proposes methods of first estimate the weights of confidences of each label source, and then learn the hypothesis. Authors provides generalization error bounds for labels sets with different cardinalities. However, no empirical evaluations are presented.

**Questions:**

- For equation (3), why this model is selected? If it is just adapted from SLV+22, then why this model is selected in SLV+22? How is it favorable than other models, if any?
- For find the Pseudo-Euclidean embedding space, is it unique? If not, how its selection and how embedding dimension would affect downstream performance? It seems like authors critique on SLV+22 can be applied as it relies on the availablity of pseudo-euclidean embeddings but does not explain how to find these.
- What are the motivations for introducing multi-view mixtures in Seciton 3.2?
- In Page 5, Line 160, what are the various moments of the outputs of LFs?
- What are intuitive reasons for the three assumptions in page 7?
- Maybe a typo in Algorithm1 Input, should be dots instead of comman in Labeling function outputs?

**Limitations:**

Authors properly addressed limitations.

**Strengths And Weaknesses:**

Strengths
- The problem setting is an important and general one, and the proposal is original.
- The writting is overall fine.

Weakness
- The title is too general. Weak supervision can also mean indirect information about labels. It would be better to emphasis the aggregation aspect of the paper.
- Motivation is overal not presented clearly. What exactly the proposed method is neccessary for dealing with structured label spaces remains unclear.
- related work is not enough. Aggregating multiple noisy labels and estimating label source confidence has been explored in fields such as crowdsourcing, at least to an empirical evaluation extent.
- Without neccessary empirical evaluation results even using synthetic data, it is difficult to judge the overall significance of the proposal.
- In Page 6, Line 197, Assumption on knowing the prior should be more emphasized. This is a strong assumption as some weakly supversied learning methods heavily depend on it, such as Positive-Unlabeled Learning with Non-Negative Risk Estimator, Kiryo et al., NeurIPS 2017.

---

> ### Author Response · Authors · 2022-08-02
> **Response to Reviewer x4kV. (Part 1)**
>
> The response is split into two comments.
>
> We appreciate the reviewer's detailed comments and feedback. We respond to each of their questions and comments, with the exception of the comments on empirical evaluation, which is addressed in the common response.
>
> * **On the title:** Indeed, "weak supervision" is a somewhat overloaded term. There is a pre-existing line of work where weak supervision refers to using multiple noisy (but cheaply-acquired) label estimates with no ground truth to build pseudolabels for training models. **Our title simply refers to this set of works. See, for example, (Ratner '18, Ratner '19, Fu '20, Shin '22), and more**. Since the work most closely related to ours (structured prediction with multiple noisy label estimates) uses "weak supervision" in this sense, we adopt this terminology as well.
>
> * **On our motivation:** We explain our motivation in the introduction (paragraphs 3 and 4), and re-state here: techniques exist for theoretically characterizing generalization performance for structured prediction models, and algorithms exist for replacing ground-truth labels in these cases with weak supervision-based noisy sources. However, prior to our work, it was not known whether weak supervision combined with structured prediction would **preserve the generalization rates**. Our work closes this gap.
>
> * **On related areas:** We agree with the reviewer---this is a large and diverse area. The point of weak supervision methods is to **subsume** techniques such as crowdsourcing. Crowdworkers can be used as the labeling functions, as can many other sources of labels, such as pretrained models, knowledge base lookups, heuristics or rules, and more.
>
> * **On needing to know the prior:** We note that in fact we do not need to know the prior---this assumption is simply for a cleaner presentation! The advantage of the tensor-product approach is that it produces an estimate of the prior for free when running the method. In fact it is possible to simply introduce the estimation error for this term into our overall bounds, which comes at the cost of less clear expressions.
>
> * **On the choice of model:** This is an excellent question. There are several reasons why we would like to work with the model in (equation 3)
>     * This is an **exponential family model** that works via the principle of maximum entropy. That is, it has the largest amount of uncertainty when choosing among valid distributions. Such uncertainty-maximizing models are often selected when modeling noise.
>     * It has **nice properties** that make it amenable to theoretical study: for example, in many settings it can be described either by the so-called canonical or mean parameters, which can be obtained in different ways and transformed into each other. We exploit this property by estimating the mean parameters using certain invariances.
>     * It **subsumes popular special cases of noise**. For example, if we are doing regression, labeling function errors reduce to zero-mean multivariate Gaussian noise; for permutations, it generalizes the popular Mallows model; for the binary case, it produces a close relative of the Ising model.
>     * Finally, it was used for weak supervision + structured prediction **in (Shin '22), enabling us to directly compare our techniques with theirs**.
>
> * **On pseudo-Euclidean embeddings:** The embeddings are **unique up to rotation**. Such rotations do not affect our approach, which is invariant to them (they preserve distances, which is the only property we require). We do not need to pick a particular orientation; it is produced by the algorithm. Similarly, the dimension of the embedding is fixed: it is essentially the number of non-zero eigenvalues of the squared distance matrix. Finally, **the algorithm to obtain pseudo-Euclidean embeddings in given in Algorithm 1 of our submitted draft**. This is in contrast to (Shin '22), where the authors suggest running MDS and then searching for another approach in cases where MDS does not produce isometric embeddings. **Algorithm 1 will always produce the desired isometric embeddings** for any finite metric space.
>
> The response is continued in the next comment.

---

> > ### Author Response · Authors · 2022-08-02
> > **Response to Reviewer x4kV. (Part 2)**
> >
> >
> > * **On multi-view mixtures:** Our motivation for using this tool is that it perfectly matches our setting. After embedding our labeling functions with pseudo-Euclidean embeddings, we can think of the resulting vectors (from different labeling functions) as multiple noisy views of the true (unobserved) vector. This immediately enables us to use tools for recovering mixtures to find the parameters of our noise model.
> >
> > * **On the moments:** We are referring to the expectations of individual labeling functions, their outer products, and, finally, their three-way tensors.
> >
> > * **On the assumptions:** These are technical conditions that lead to clean expressions; all of them can be relaxed at the cost of less revealing formulas. More concretely, Assumption 4 is used to avoid function $F$ having multiple minimizers or being flat around a minimizer. This is used to bound the distance between the minimizers from the true and noisy labels, respectively. Assumption 2 ensures that the noise model P is not ill-conditioned, and Assumption 3 guarantees that the random variables under consideration are bounded.
> >
> > * **Typo in Algorithm 1:** Thank you for spotting this! We have corrected it.

---

> > > ### Comment · Reviewer_x4kV · 2022-08-07
> > > **Reply**
> > >
> > > I would like to thank authors hard work on addressing my concerns and continuously improving the paper. My main concerns have been addressed properly.
> > > - Unnecessariness of assuming the prior is known.
> > > - The favorable properties of the chosen exponential family model.
> > > - The uniqueness of the pseudo-Euclidean embedding and the property of Algorithm 1.
> > > I also appreciate that authors append empirical evaluation results with detailed python codes. I have raised my scores.

---

### Official Review · Reviewer_tEkM · 2022-07-11

**Rating:** 7
**Confidence:** 3
**Soundness:** 4 excellent
**Presentation:** 4 excellent
**Contribution:** 3 good

**Summary:**

Most prior work on weak supervision with theoretical guarantees has focused on the binary or multi-way classification settings. This paper provides theoretical guarantees for weak supervision in a broader range of label settings like finite metric spaces, for example, rankings (where the label for every data point is an ordering of {1,2,..., k} - if there are k categories, with the Kendall tau metric), and manifold-valued label spaces. In particular, for the case where the labels belong to a finite metric space, they provide an algorithm for pseudo-label construction using pseudo-Euclidean embeddings and tensor decompositions, which they claim are new to weak supervision, and for the case of labels in constant-curvature Riemannian manifolds, they provide an invariant which they then use for noise rate estimation of the labeling functions. Using the pseudo-labels formed by their procedures in down-stream models from prior work [CRR16, RCMR18], they provide generalization error bounds for both these cases.

CRR16 - Carlo Ciliberto, Lorenzo Rosasco, and Alessandro Rudi. A consistent regularization approach for structured prediction. In Advances in Neural Information Processing Systems 30 (NIPS 2016), volume 30, 2016.
RCMR18 - Alessandro Rudi, Carlo Ciliberto, GianMaria Marconi, and Lorenzo Rosasco. Manifold structured prediction. In Advances in Neural Information Processing Systems 32 (NeurIPS 2018), volume 32, 2018.

**Questions:**

The authors claim that tensor decompositions are new to weak-supervision but [SLV+22] mentions “We require an estimate of the prior p on the label y; there are techniques do so (Anandkumar et al., 2014; Ratner et al., 2019); we tacitly assume we have access to it.” on page 5. I have not read [SLV+22] very carefully but it seems like they were the first to notice that tensor decomposition based approaches can be used for weak supervision.


**Limitations:**

This is primarily a theoretical paper and so the authors have chosen N/A for checklist item 3 [Did you discuss any potential negative societal impacts of your work].

**Strengths And Weaknesses:**

Originality: I am not an expert in this area so I am not entirely sure about other related work. From what I can read from the paper, it indeed does seem like there is not much prior work on weak supervision for structured prediction, which clearly is an interesting area. The only other paper which they say that has studied WS for structured prediction is a recent ICLR 2022 paper [SLV+22], which seems to lean more on the empirical side.

Quality: The submission is technically sound. All claims are well-supported with proofs.

Clarity: The submission is clearly written and well-organized.

Significance: Yes, the problems being studied are quite significant. Structured prediction settings like rankings are very well studied in the literature and they also seem to have a lot of practical applications so weak-supervision algorithms for them can have a good impact.

SLV+22 - Changho Shin, Winfred Li, Harit Vishwakarma, Nicholas Carl Roberts, and Frederic Sala. Universalizing weak supervision. In International Conference on Learning Representations, 2022.

---

> ### Author Response · Authors · 2022-08-02
> **Response to Reviewer tEkM**
>
> We thank the reviewer for their kind words and praise for our work. We answer the reviewer's question:
>
> * **On the use of tensor decompositions in weak supervision:** The reviewer notes that other works in this area, including (Shin '22) describe the use of tensor decompositions. Indeed, this is the case---**but only for prior estimation**. In fact the idea of estimating mixture priors using tensor decompositions is much older, and (Shin '22) refers to this. What these papers do not do, however, is to learn the parameters of the noise model (equation 3), or any of its specialized versions, using tensor decompositions. This is the focus of our paper. We appreciate the reviewer asking, as this distinction is indeed important.

---

### Official Review · Reviewer_J1JM · 2022-07-17

**Rating:** 6
**Confidence:** 3
**Soundness:** 3 good
**Presentation:** 2 fair
**Contribution:** 3 good

**Summary:**

This article proposes a weak supervision technique for structured prediction, with labels taking values in finite metric spaces and constant-curvature Riemannian manifolds, and assuming an exponential-family-based probabilistic label model for the weak labels parametrized by the accuracy potentials.

For finite metric spaces, the authors incorporate structure into the label model by first computing pseudo-Euclidean embeddings of the labels, making use of native distance in the label space.  Similar to [1], a tensor decomposition algorithm is used on the pseudo-embeddings to obtain estimators for the mean parameters of the label model.  Pseudo-labels are created for each data point by sampling from conditional distribution of the latent truth conditional on the output of labeling functions.  The authors then proceed in a similar way to [2], where the label space distance is is a weighted (according to the estimated conditional distribution) aggregation of distances to the weak labels.  The authors derived an upper bound for the parameter recovery error for both high and low noise regimes.  An upper bound for generalization error is also derived and shown to have an extra (compared to the bound in [2]) additive factor which is dependent on the support size of the label space and embedding dimension size.

For constant-curvature Riemannian manifolds, a generalization of the triplet method (similar to [3,4]) which leverages the law of cosines is proposed, which allows for a closed form estimator for the distance between weak label and latent truth.  This estimator is shown to be consistent. The authors then proceed in a similar to way to [2], where the label space distance is a minimized convex combination of distances to the weak labels.

The paper focuses on the theoretical aspects of the algorithm, without any experimentation.

[1] Tensor decompositions for learning latent variable models. Anandkumar et al.

[2] A consistent regularization approach for structured prediction. Ciliberto et al.

[3] Fast and Three-rious: Speeding Up Weak Supervision with Triplet Methods. Fu, Chen et al.

[4] Evaluating the Crowd with Confidence. Joglekar et al.



**Questions:**

1. Why is tensor decomposition ([1]) chosen for parameter estimation? Would other approaches work?  Presumably you would want the method to have guarantees so that you can derive an analog of Theorem 1.

2. Can the authors please comment on/reconcile the differences between the $O(n^{-1/4})$ generalization error proposed to the $O(n^{-1/2})$ in [2]?

3. Can the multi-view mixture model handle correlations between the views?  That might be how the correlation potential can be handled in the probabilistic model.

[1] Tensor decompositions for learning latent variable models. Anandkumar et al.


[2] Fast and Three-rious: Speeding Up Weak Supervision with Triplet Methods. Fu, Chen et al.


**Limitations:**

Adequately addressed

**Strengths And Weaknesses:**

Strengths:

1) Adapting of the guaranteed general approach to structured prediction in [1] to the case of weak supervision is a natural goal.  At a high level, the general approach is simple to understand and does not change very much -- we only need to swap the label space distance term to a weighted combination of the distances to the weak labels.
2) The design choice of using pseudo-Euclidean embeddings seems to be very appropriate, to accomodate general dissimilarity matrix from general finite metric structured spaces.
3) Parameter recovery error and generation error bounds seem to be rigorously derived.  The desciption of practical consequences, for example, on Line 186 ("This leads to tradeoff in selecting the appropriate embedding dimension") and on Lines 245-246 ("Thus we see that we only pay an extra additive factor..."), are highly appreciated.

Weaknesses:

1) The choice of noise model (Equation 3) seems to exclude correlation potentials, which may limit the scope of practicality.
2) It is not clear how the bounds presented compare to bounds in prior works for binary/multiclass labels.
3) It is not clear why tensor decomposition from [4] is chosen for parameter estimation.
4) Some presentation aspects can be improved:
- Line 36, it is implied that all WS methods for binary classification yields models with generalization guarantees, but this is not the case. See [2] for other weak supervision methods.
- Line 106, there should a space in "small.In"
- Line 109, should there be $\theta_a$ there? The sentence seems awkward.
- Line 110, "and $\theta_a = \infty$, so that $\theta_a = y$ seems awkward.
- Line 143, it may not be clear for readers who are not familiar with pseudo-Euclidean embedding, why the pseudo-Euclidean embedding of the example is 3.  Moreover, I do not think there is enough citation on pseudo-Euclidean embedding and its computation in Algorithm 1.
- Line 145, it might be clear to write "embeddings of $\lambda_{a}, y$ respectively".
- In Equation 4, instead of a period, it should be a comma at the end of the line.
- Line 200, there seems to be something missing after "where $\tilde{z}_i$".
- Line 243, "as in the case of access toclean labels" needs to be fixed.
- Line 272, "Note that the finite metric-space case is insufficient: the support is infinite".  This sentence is hard to understand.
- Lines 325,326, "it is possible to do for..." is ungrammatical.
- Line 290, I find "C_0 reflects the pushforward of concentration between the distributions ..." dubious -- I have never seen "pushforward" used for "concentration" before.
- Appendix E, Definition 1. Is this missing "triangle inequality" and "need not be distinguishable"?
5. It would have been interesting for weak supervision practitioners to see some experiments to study the empirical performance, perhaps comparing against the approach proposed in [3].

[1] A consistent regularization approach for structured prediction. Ciliberto et al.

[2] WRENCH: A Comprehensive Benchmark for Weak Supervision. Zhang et al.

[3] Universalizing weak supervision. Shin et al.

[4] Tensor decompositions for learning latent variable models. Anandkumar et al.

---

> ### Author Response · Authors · 2022-08-02
> **Response to Reviewer J1JM**
>
> We appreciate the reviewer's excellent feedback, close reading of our paper, and suggestions that have benefitted our work. We start with the main questions and address smaller ones and suggestions afterwards.
>
> * **On the choice of using tensor decomposition in our algorithm:** Our choice is based on two requirements. First, we need a technique that can provide *consistent or nearly-consistent estimates* of the parameters in the noise model. Second, we need a technique that can handle *any* finite metric space. Techniques like the one introduced in (Fu '20) handle the first---but do not work for generic finite metric spaces, only binary labels and certain sequences. Techniques like the one in (Shin '22) handle any metric space---but only have consistency guarantees in highly restrictive settings (e.g., it requires an isometric embedding, that the distribution over the resulting embeddings is isomorphic to certain distributions, the true label only takes on two values).
>
>     Combined with pseudo-Euclidean embeddings, tensor decomposition algorithms **meet both requirements**. We do not know of any other approach that can simultaneously accomplish both. This motivates our use of these techniques (and their combination) and forms one of the major results of our work.
>
> * **On the generalization error rates:** The reviewer asks about how to compare the rates we have in this paper, $O(n^{-1/4})$, versus those from Fu '20, which are $O(n^{-1/2})$. Simply put, **the rates are a function of the noiseless model and its rates**. For the type of binary classification problems Fu '20 is concerned with, the noiseless rate is $O(n^{-1/2})$. The noiseless version of the structured prediction estimator we work with (Ciliberto '16) has a rate of $O(n^{-1/4})$. We note that these are not always directly comparable, since they are measuring different quantities, e.g., the structured prediction estimator's generalization definition measures a certain average distance.
>
>     How does weak supervision fit in? Adding noise makes it more difficult to obtain the same generalization performance (we must estimate noise parameters, obtain high-quality pseudolabels, and deal with their remaining noise in these pseudolabels when training) , so our goal is to **match the noiseless rate**. Working in the binary (and multiclass) settings, Fu '20 showed that the rate is preserved: weak supervision still allows for the $O(n^{-1/2})$ rate. Our goal was to provide a matching result for structured prediction: we were able to do so: $O(n^{-1/4})$ preserves the rate of the Ciliberto '16 structured prediction estimator. In this sense, we produce the same type of result as (Fu '20)---but for a much more difficult problem.
>
> * **On handling correlations:** This is an excellent question. **Our approach works with correlation potentials**, and our presentation in (3) was for simplicity. We have updated the draft to include a set of correlations $E$, which our approach can handle for a wide variety of scenarios. Specifically, what is required is that the labeling functions (LFs) form at least three clusters with no correlation potentials between cross-cluster pairs. The sizes of these clusters can be arbitrary and there can be any number of correlations between pairs of LFs within clusters. For example, we can have one cluster with $m-2$ LFs that are all mutually correlated along with two other clusters with $1$ LF each. In other words, the number of correlations can be very large, even $O(m^2)$, and we can still recover.
>
>     The reason for this requirement is that as long as we select triplets of LFs that are conditionally independent, we can apply the triplets approach in Algorithm 1 to recover their parameter $\theta_a$. Learning the correlations is also easily done, as here we can directly estimate the pairwise mean parameter. An important consideration is to know which LFs are in which cluster, which requires structure learning. We note that there is an easy way around this as well---select many random triplets when computing the parameter $\theta_a$ for each $1 \leq a \leq m$ and take the median of the resulting estimates. (Chen '21) studies such median estimators and these results apply to our work as well; they show that even without knowledge of the clusters, we are likely to recover a result coming from conditionally independent views as long as such views exist in the first place.
>
> * **On empirical evaluations:** We refer to the common response above.
>
> * **On presentation issues:** We appreciate the reviewer's careful reading and we have taken all of these suggestions. We only comment below on cases where clarification is useful.
>     * Indeed, not all WS binary methods have generalization guarantees---just some (i.e., Ratner '19, Fu '20).
>     * On our example of a useful pseudo-Euclidean embedding: we provide additional context in the appendix.
>     * Indeed, the triangle inequality holds in pseudometric spaces.

---

> > ### Comment · Reviewer_J1JM · 2022-08-07
> > **Reply to authors**
> >
> > I would like to thank the author(s) for a thorough response and including the code.  This is an important practical problem which could induce further research projects, and having a reproducible code would allow future works to study this problem more efficiently.

---

### Author Response · Authors · 2022-08-02
**Common Response to All Reviewers**

### Common Response to All Reviewers

We thank all the reviewers for their kind comments and detailed feedback. We believe their reviews have substantially improved our draft. All of the reviewers noted the strength of our work, the importance of the problem setting, and the originality of our techniques. In addition, we have made improvements since submission and updated our draft; we mark changes that correspond to reviewer comments in blue in the updated draft. We first provide a common response before answering each reviewer in-depth. The common response identifies and comments on two threads:

* **Experiments**: The reviewers appreciated our strong theoretical results, but mentioned that experiments illustrating these would further improve the work. We agree. Since submission, **we have added experimental results** that verify our key claims. These can be found in the appendix of our updated draft, or directly accessed [here](https://imgur.com/a/GlenfJ9). We have also uploaded the code and the appendix as supplementary material in a single zip file.

    These experiments **confirm the two central claims in our paper**. First, they show that, as expected, the estimators we propose (using the pseudo-Euclidean embeddings + tensor decomposition in the finite metric-space case or using the law of cosines in the manifold case) **are (nearly) consistent, as derived in our theoretical results**. Second, we compare directly to the competitor methods in (Shin '22); we observed that our approaches **produced significantly stronger performance, both in terms of parameter recovery and downstream structured prediction task**.

* **Background on pseudo-Euclidean embeddings:** Several reviewers mentioned that while our use of these embeddings is original, more background on these should be included. We agree and have added additional comments and explanations in our updated draft. We provide additional context in the body and the appendix. In short, embeddings of squared distances into Euclidean space via multidimensional scaling (MDS) are isometric as long as the matrix used for the embeddings is p.s.d.---but this is often not the case. **Pseudo-Euclidean spaces have two components (a positive and negative component) which can then accomodate both p.s.d and negative definite terms---enabling isometric embeddings for any finite metric space**. The "cost" of using such spaces is that they have non-standard properties: for example, two points can be distinct but still have distance zero. However, since we only use these spaces to represent a set of points with fixed distances (coming from our metric space), we never suffer from such disadvantages, while we get to exploit isometric embeddings.


### References
(Chen et al. 2021) Comparing the Value of Labeled and Unlabeled Data in Method-of-Moments Latent Variable Estimation. AISTATS, 2021.
(Ciliberto et al. 2016) A Consistent Regularization Approach for Structured Prediction. NIPS, 2016.
(Fu et al. 2020) Fast and three-rious: Speeding up weak supervision with triplet methods. ICML, 2020.
(Ratner et al. 2019) Training Complex Models with Multi-task Weak Supervision. AAAI, 2019.
(Ratner et al. 2018) Snorkel: Rapid Training Data Creation with Weak Supervision. VLDB, 2018.
(Shin et al. 2022) Universalizing Weak Supervision. ICLR, 2022.

---

### Author Response · Authors · 2022-08-06
**Thank You Again and Further Discussion**

We wish to thank the reviewers for their feedback one more time. We enjoyed responding and believe that it has made our paper even stronger.

We would love to continue the conversation! Only a few days remain for the discussion period, so we would like to ask if there are further questions, clarifications, or extensions to bring up.

---

### Meta-Review · Area_Chair_DeGW · 2022-08-28

**Recommendation:** Accept
**Confidence:** Less certain

**Metareview:**

The paper gives theoretical guarantees for weak supervision for more general problems than binary classification. The reviewers were all positive about this work and felt this paper introduces new techniques to this space that may be useful for other problems as well.

**Award:**

No

---

### Decision · Program_Chairs · 2022-09-14

Accept